# Nucleotide depletion reveals the impaired ribosome biogenesis checkpoint as a barrier against DNA damage

Joffrey Pelletier[1,*,†,‡] , Ferran Riaño-Canalias[1,†], Eugènia Almacellas[1] , Caroline Mauvezin[1] , Sara Samino[2,3], Sonia Feu[4], Sandra Menoyo[1], Ana Domostegui[1], Marta Garcia-Cajide[1], Ramon Salazar[1,5], Constanza Cortés[6], Ricard Marcos[6], Albert Tauler[1,7], Oscar Yanes[2,3], Neus Agell[4] , Sara C Kozma[1], Antonio Gentilella[1,7] & George Thomas[1,8,**] 

## Abstract

Many oncogenes enhance nucleotide usage to increase ribosome content, DNA replication, and cell proliferation, but in parallel trigger p53 activation. Both the impaired ribosome biogenesis checkpoint (IRBC) and the DNA damage response (DDR) have been implicated in p53 activation following nucleotide depletion. However, it is difficult to reconcile the two checkpoints operating together, as the IRBC induces p21-mediated G1 arrest, whereas the DDR requires that cells enter S phase. Gradual inhibition of inosine monophosphate dehydrogenase (IMPDH), an enzyme required for *de novo* GMP synthesis, reveals a hierarchical organization of these two checkpoints. We find that the IRBC is the primary nucleotide sensor, but increased IMPDH inhibition leads to p21 degradation, compromising IRBC-mediated G1 arrest and allowing S phase entry and DDR activation. Disruption of the IRBC alone is sufficient to elicit the DDR, which is strongly enhanced by IMPDH inhibition, suggesting that the IRBC acts as a barrier against genomic instability.

**Keywords** IMPDH; IRBC; nucleotides; p21; p53
**Subject Categories** Cell Cycle; DNA Replication, Recombination & Repair; Signal Transduction
The EMBO Journal (2020) 39: e103838

## Introduction

The reprogramming of metabolic networks is a hallmark of cancer, which is initiated by the activation of oncogenes or the loss of tumor suppressors, acting to drive and sustain tumor cell proliferation (Hanahan & Weinberg, 2011). One of the most studied oncogenes is c-Myc, whose dysregulation or amplification has been implicated in > 70% of all cancers (Murphy *et al*, 2008). c-Myc is a master regulator of both cancer initiation and progression (Stine *et al*, 2015), through its ability to control the transcription of a wide range of genes (Kress *et al*, 2015). A critical set of these genes promote anabolic metabolism, including the synthesis of lipids, amino acids, and nucleotides, required for tumor cell proliferation (Kress *et al*, 2015; Stine *et al*, 2015). However, c-Myc appears to be distinct from other oncogenes, many of which fail to upregulate anabolic genes involved in nucleotide metabolism (Bester *et al*, 2011; Aird *et al*, 2013), including Rb-E2F (Bester *et al*, 2011) and RAS-RAF (Aird *et al*, 2013). In those cases, oncogene activation can lead to a decrease in nucleotide pools, replicative stress, and genomic instability, argued to be an initial step in tumorigenesis (Bester *et al*, 2011; Aird *et al*, 2013). Indeed, ectopic expression of c-Myc in Rb-E2F tumorigenic model re-establishes the nucleotide pools (Bester *et al*, 2011), an effect partially recapitulated by the ectopic expression of its target gene *IMPDH2* (Mannava *et al*, 2008). Both IMPDH1 and 2 catalyze the rate limiting step in GMP synthesis, the oxidation of inosine monophosphate (IMP) to xanthosine monophosphate (XMP) (Huang *et al*, 2018). The identification of the IMPDHs as key targets of c-Myc has opened a potential therapeutic window for

1 Laboratory of Cancer Metabolism, ONCOBELL Program, Institut d'Investigació Biomèdica de Bellvitge—IDIBELL, L'Hospitalet de Llobregat, Spain
2 Metabolomics Platform, IISPV & University Rovira i Virgili, Tarragona, Spain
3 Spanish Biomedical Research Center in Diabetes and Associated Metabolic Disorders (CIBERDEM), Madrid, Spain
4 Department of Biomedicine, Faculty of Medicine, IDIBAPS Biomedical Research Institute, Hospital Clinic, University of Barcelona, Barcelona, Spain
5 Catalan Institute of Oncology (ICO), Barcelona, Spain
6 Department of Genetics and Microbiology, Faculty of Biosciences, Autonomous University of Barcelona, Barcelona, Spain
7 Department of Biochemistry and Physiology, Faculty of Pharmacy, University of Barcelona, Barcelona, Spain
8 Department of Physiological Sciences, Faculty of Medicine and Health Science, University of Barcelona, Barcelona, Spain
   *Corresponding author. Tel: +34 260 7138; E-mail: jpelletier@idibell.cat
   **Corresponding author. Tel: +34 260 7138; E-mail: gthomas@idibell.cat
   †These authors contributed equally to this work
   ‡Present address: Colorectal Cancer Laboratory, Institute for Research in Biomedicine (IRB Barcelona), Barcelona, Spain

synthetic lethality (Stine *et al*, 2015), a concept supported by recent studies showing that a subset of small-cell lung carcinomas (SCLCs), characterized by c-Myc and IMPDH upregulation, are highly vulnerable to IMPDH inhibitors (Huang *et al*, 2018).

Although nucleotides are essential for DNA replication and repair (Tong *et al*, 2009; Aird *et al*, 2013), they are largely consumed in the production of ribosomes, with ribosomal RNA (rRNA) representing ~80% of the nucleic acids of the cell and ~ 15% of its biomass (Pelletier *et al*, 2018). The demand for nucleotides is further exacerbated in tumors driven by c-Myc, which coordinates the enhanced transcription of genes required for the hyperactivation of ribosome biogenesis, needed to increase protein synthetic capacity, cell proliferation, and ultimately malignancy (Santagata *et al*, 2013). Given the dependency of such tumors on ribosome biogenesis (Barna *et al*, 2008; van Riggelen *et al*, 2010), a search for novel agents that specifically target this process has been initiated (Bywater *et al*, 2012; Pelletier *et al*, 2018). Importantly, insults to ribosome biogenesis, including c-Myc-induced oncogenic stress (Macias *et al*, 2010; Morcelle *et al*, 2019), also trigger a p53-mediated cell-cycle checkpoint, recently termed the impaired ribosome biogenesis checkpoint (IRBC) (Gentilella *et al*, 2017; Pelletier *et al*, 2018). The IRBC is mediated by a nascent pre-ribosomal complex containing RPL5 (or uL18), RPL11 (or uL5), and 5S rRNA, which upon insults to ribosome biogenesis, is redirected from its assembly into 60S ribosomes to the binding and inhibition of the p53-E3-ubiquitin ligase HDM2 (or MDM2) (Donati *et al*, 2013), leading to p53 stabilization (Kubbutat *et al*, 1997; Pelletier *et al*, 2018). In most cases, p53 stabilization by the IRBC leads to G1 cell-cycle arrest, largely mediated by the cyclin-dependent kinase inhibitor 1 (CDKN1A) or p21. This mechanism is distinct from that involving replicative stress and the DNA Damage Response (DDR), where phosphorylation of either HDM2 or p53 prevents their interaction, stabilizing p53 (Bode & Dong, 2004), independent of the IRBC (Macias *et al*, 2010).

Because of the critical role of IMPDHs in proliferating tumor cells (Liu *et al*, 2008; Mannava *et al*, 2008), a number of studies have focused on the development of antimetabolites, which inhibit their function (Stine *et al*, 2015), including mycophenolic acid (MPA), mizoribine, and AVN-944. These agents are catalytic non-competitive inhibitors of IMPDH, which unlike nucleoside analogs that produce chromosome breaks or inhibit DNA repair enzymes, do not incorporate into the DNA (Allison & Eugui, 2000). However, the underlying mechanisms by which they lead to p53 stabilization are not clearly understood. Initial studies suggested that IMPDH inhibitors activate a reversible p53-dependent cell-cycle checkpoint (Linke *et al*, 1996), which was later attributed to the IRBC (Sun *et al*, 2008). However, others have reported that IMPDH inhibition leads to replicative stress and the DDR (Liu *et al*, 2008). This seeming contradiction was recently highlighted in studies of two human tumor types, in SCLCs characterized by the low expression of the Achaete-scute homolog-1 (ASCL1) and the upregulation of c-Myc (Huang *et al*, 2018) and in tuberous sclerosis complex 2 (TSC2)-deficient kidney tumors, where increased rRNA synthesis is driven by constitutive mTOR signaling (Valvezan *et al*, 2017). Both are sensitive to IMPDH inhibitors, but apparently through different mechanisms: the first through the inhibition of ribosome biogenesis (Huang *et al*, 2018) and the second through DDR (Valvezan *et al*, 2017). The differences in these findings are not clearly understood, as is the apparent incompatibility between the induction of the IRBC, leading to G1 arrest in a p53 wild-type setting (Sun *et al*, 2008), and replicative stress.

Given the wide use of nucleotide synthesis inhibitors in cancer therapy, it is important to understand the cellular checkpoints induced by these anticancer agents. Here, we set out to address the contribution of the IRBC and replicative stress in driving p53 stabilization upon guanine nucleotide imbalance in sporadic colorectal cancer (sCRC) cell lines, as almost all CRCs are initiated by c-Myc dysregulation (TCGA, 2012) and are addicted to hyperactive ribosome biogenesis (Pelletier *et al*, 2018). We find that a reduction of guanine nucleotide pools first impairs ribosome biogenesis, leading to the induction of the IRBC, p53 stabilization, and G1 arrest. However, if nucleotide depletion becomes more severe, cells enter S phase and encounter replicative stress, which is paralleled by the decreased expression of p21. Although the levels of nascent 5S rRNA, an essential component of the IRBC complex decrease under these conditions, we find that this is not sufficient to impinge on complex formation nor the transcriptional induction of p21. Instead reduced p21 protein levels appear to be attributed to enhanced proteasome degradation. Moreover, S phase entry is further enhanced in MPA-treated $p21^{-/-}$ cells, with ectopic expression of p21 reversing this response. Consistent with this observation, the downregulation of the IRBC complex, reducing p21 transcription, facilitates MPA-induced S phase entry despite low guanine nucleotides levels, leading to further DNA damage. Unexpectedly, loss of the IRBC alone triggered DNA damage. These findings demonstrate that guanine nucleotide levels differentially control two distinct p53-checkpoints in a hierarchical manner and that the IRBC acts as a barrier against DNA damage.

# Results

### Distinct mechanisms mediate p53 activation in response to decreasing guanine nucleotides

To identify the molecular mechanisms by which guanine nucleotide depletion regulates p53, we carried out a dose-response analysis with the IMPDH non-competitive catalytic inhibitor, MPA. As reported earlier in U2OS osteosarcoma cells (Sun *et al*, 2008), 24 h treatment of HCT116 cells with MPA leads to a dose-dependent increase in p53 protein levels (Fig 1A). Similar maximal levels of p53 were observed following treatment with 5 nM Actinomycin D (ActD) (Fig 1A), which selectively inhibits Pol I-dependent rRNA transcription (Perry, 1963), leading to activation of the IRBC (Donati *et al*, 2013). However, as the concentration of MPA increased, the amount of p21 protein was reduced (Fig 1A and Appendix Fig S1A). Moreover, concomitant with the decrease in p21 protein levels, the phosphorylation of p53 S15, Chk1 S345, and BRCA1 S1524 was increased, markers of replicative stress and ATR activation, absent in cells treated with 1 μM MPA or ActD (Fig 1A). However, at these time points, MPA does not lead to the phosphorylation of the ATM targets Chk2 T68 or H2AX S139 (γ-H2AX), nor increases the number of γ-H2AX or 53BP1 foci, indicating that DNA double-strand breaks (DSBs) are not a primary consequence of GMP depletion (Fig 1A and Appendix Fig S1B–D, respectively). Similar findings were obtained with the non-competitive inhibitor of IMPDH, AVN944 (Appendix Fig S1E). In contrast, inhibition of topoisomerase II by

etoposide, which generates DNA-double strand breaks, led to an increase in Chk2 T68 phosphorylation and γ-H2AX (Appendix Fig S1F). These results suggest that at lower concentrations of IMPDH inhibitors the IRBC is activated and p53 stabilized, whereas at higher concentrations ATR and p53 phosphorylation are induced.

Given the differential effects of IMPDH inhibitors on p53 stabilization and phosphorylation, we asked whether these differences were associated with changes in guanine nucleotides pools, as measured by liquid chromatography-mass spectrometry (LC-MS). Following 24 h treatment with 1 or 10 μM MPA, we observed a dose-dependent drop in all guanine ribonucleotides, whereas dGTP levels only decreased significantly in 10 μM MPA-treated cells (Fig 1B). Moreover, IMPDH inhibition leads to the accumulation of the substrate IMP (Appendix Fig S1G) but has no effect on adenine nucleotide levels (Appendix Fig S1H), pyrimidine nucleotide UTP levels (Appendix Fig S1I), or on the $NAD^+$/NADH redox state, despite $NAD^+$ being an IMPDH coenzyme (Appendix Fig S1J). The differential effects of increasing concentrations of MPA on guanine nucleotide levels appear to be compatible with IMPDH inhibition eliciting distinct effects on ribosome biogenesis and replicative stress.

### The induction of p53 by MPA is selectively mediated by IMPDH inhibition

To ensure that the effects of MPA on p53 were mediated by IMPDH inhibition, we depleted cells of IMPDH1 and/or IMPDH2, the dominant isoform in most mammalian tissues (Carr *et al*, 1993; Senda & Natsumeda, 1994), and measured the responses above. We first generated a stable HCT116 cell line expressing a tetracycline (Tet)-inducible miR30 based shRNA (Zuber *et al*, 2011) against IMPDH2 (IM2$^{iKD}$). The IM2$^{iKD}$ parental cell line was then transfected with either a non-targeting (NT) or IMPDH1 shRNA, to generate the IM2$^{iKD}$-shNT and IM2$^{iKD}$–shIM1 stable cell lines. In IM2$^{iKD}$-shNT cells, tetracycline treatment did not affect IMPDH1 mRNA levels (Fig 1C, left panel), whereas IMPDH2 mRNA levels were reduced (Fig 1C, right panel). In IM2$^{iKD}$–shIM1 cells, IMPDH1 mRNA levels were decreased (Fig 1C, left panel), with tetracycline treatment inducing a reduction in IMPDH2 mRNA levels (Fig 1C, right panel). The shRNA targeting of IMPDH1 had no effect on total IMPDH (see Materials and Methods), p53 or p21 protein levels, nor on Chk1 S345 phosphorylation (Fig 1D). Despite IMPDH2 depletion decreasing the levels of IMPDH2 and total IMPDH protein, while inducing p53 and p21 protein, it had no effect on Chk1 S345 phosphorylation (Fig 1D). Similar results were obtained with an independent shRNA sequence against IMPDH2 (Appendix Fig S1K and L). These findings argue that IMPDH2 is the predominant isoenzyme in HCT116 cells and that its depletion induces p53, but apparently independent of replicative stress.

The failure of IMPDH depletion to induce Chk1 phosphorylation (Fig 1D) suggested that the decrease in IMPDH levels was not severe enough to lower nucleotide levels to the level achieved with MPA (Fig 1B). In agreement with this observation, addition of 10 μM MPA in cells depleted of IMPDH2 led to an increase in Chk1 S345 phosphorylation (Appendix Fig S1M) and a further decrease in guanine nucleotide levels (Appendix Fig S1N). We also assessed potential off-target effects of MPA by supplementing cells with exogenous guanosine, through which the purine salvage pathway

can regenerate nucleotides from degradative intermediates. The addition of exogenous guanosine bypassed the effects of MPA on guanine nucleotide levels (Fig 1B), completely preventing the induction of p53, p21, and Chk1 S345 phosphorylation by 10 μM MPA (Fig 1E), with none of the other three nucleosides showing a similar effect on p53 (Appendix Fig S1O). These results support the notion that as guanine nucleotide levels decrease distinct responses are activated, which converge on p53.

### The IRBC and ATR contribute distinctly to the regulation of p53

The high demand of ribonucleotides for ribosome biogenesis (Pelletier *et al*, 2018) supports the concept that the p53 stabilization, by low concentrations of IMPDH inhibitors, is mediated by the IRBC. To test this hypothesis, we pretreated cells with a siRNA NT or a siRNA against RPL11 (siRPL11), an essential component of the IRBC (Pelletier *et al*, 2018). The results show that the upregulation of p53 by 1 or 10 μM MPA is reduced by depletion of RPL11; however, the levels of p53 are significantly higher in cells treated with 10 μM MPA (Fig 2A), despite an equivalent decrease in RPL11 mRNA levels (Appendix Fig S2A). To ensure that the second input to p53 is independent of the IRBC, we depleted cells of RPL7a (or eL8), an essential RP of the 60S ribosomal subunit whose depletion leads to the activation of the IRBC (Fumagalli *et al*, 2012), or co-depleted RPL7a and RPL11 (Fig 2B and Appendix Fig S2B), followed by treatment with 10 μM MPA for 6 h. The results show that depletion of RPL11 alone has no effect on p53 or p21 levels, whereas the strong induction of both responses following depletion of RPL7a is completely reversed by co-depletion of RPL11 (Fig 2B). However, p53 upregulation by MPA in control or RPL7a-depleted cells is only partially reversed by RPL11 co-depletion (Fig 2B). These findings support the role of the IRBC in the primary response to nucleotide depletion, and that there is a distinct secondary input to p53 when the inhibition of GMP synthesis is more severe.

As ATR-mediated p53 S15 phosphorylation (Fig 1A) impairs its interaction with HDM2 (Bode & Dong, 2004), the ATR pathway is the likely candidate for the induction of the second p53 checkpoint by MPA. Although selective ATR inhibitors block kinase activity, ATR protects the stability of replicative forks, such that its inhibition leads to the collapse of replicative forks and to ATM-mediated p53 S15 phosphorylation, which is amplified by nucleotide shortage (Chanoux *et al*, 2009; Toledo *et al*, 2013). Consistent with this observation, the ATR inhibitor (VE-821, ApexBio) strongly inhibited Chk1 S345 phosphorylation, but induced the phosphorylation of p53 S15, Chk2 T68, and γ-H2AX (Appendix Fig S2C), markers of ATM activation. The increase in p53 S15 and Chk2 T68 phosphorylation was suppressed by the ATM inhibitor (KU-55933, Sigma-Aldrich), whereas that of γ-H2AX was unaffected, most likely attributed to the activation of DNA-dependent protein kinase (Stiff *et al*, 2004). Analysis of the combination of ATR and ATM inhibitors on p53 activation by 10 μM MPA showed only a partial inhibition of p53 stabilization (Fig 2C and Appendix Fig S2C), which was also the case for RPL11 depletion (Fig 2C and Appendix Fig S2D). However, ATR/ATM inhibition in cells depleted of RPL11 rescued p53 to almost basal levels (Fig 2C), consistent with a milder lesion in guanine nucleotide synthesis inducing the IRBC, whereas a more severe lesion causes induction of ATR and p53 phosphorylation.

**A**

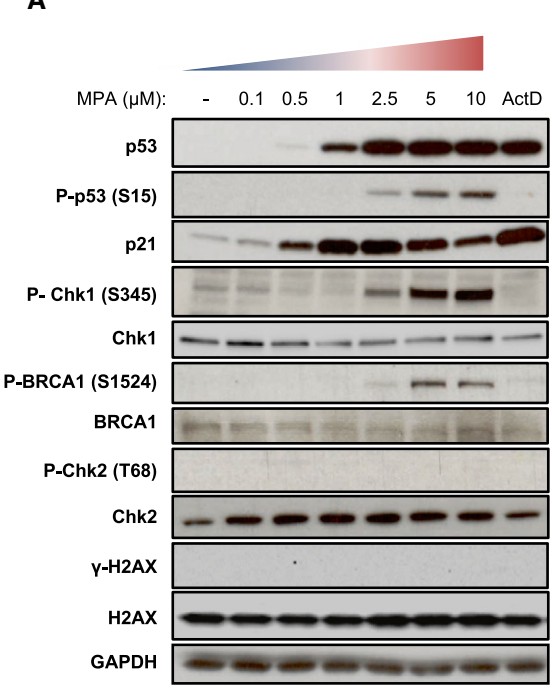

**B**

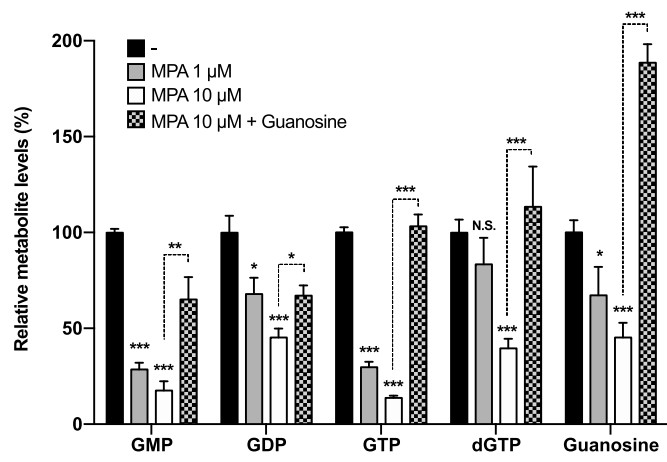

**C**

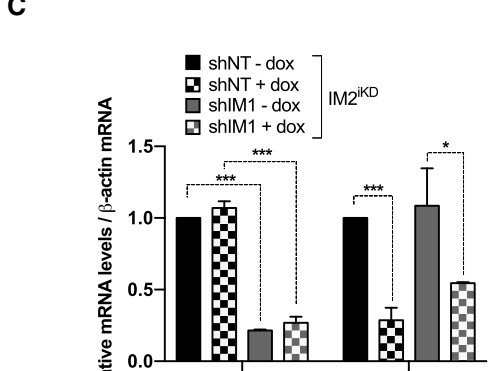

**D**

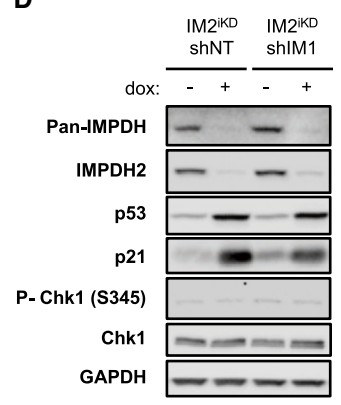

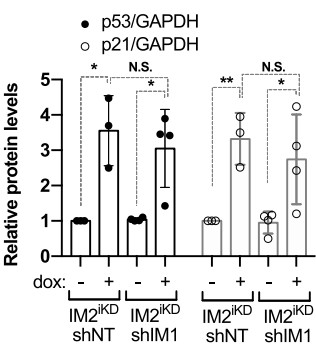

**E**

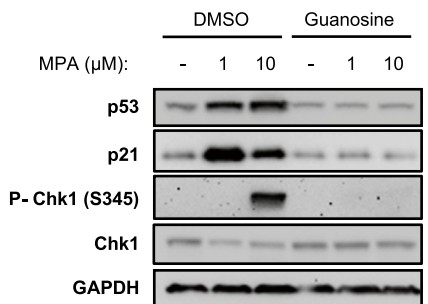

Figure 1.

◀

**Figure 1.  Inhibition of guanosine nucleotide synthesis elicits distinct p53 responses.**

A Western blots in HCT116 cells treated for 24 h with increasing concentrations of MPA or 5 nM Actinomycin D (ActD). GAPDH served as a loading control.

B HCT116 cells were treated for 24 h with vehicle alone (−), 1 μM MPA, 10 μM MPA, or the combination of 10 μM MPA with 400 μM guanosine. Guanylate nucleotide levels were measured by LC-MS and normalized to protein content. Mean ± SEM is representative of 3 independent experiments carried out in triplicate.

C, D Parental HCT116 cells expressing a stable tetracycline-inducible shRNA against IMPDH2 (IM2[iKD]) and either a stable non-targeting (NT) shRNA (IM2[iKD]-shNT) or an shRNA targeting IMPDH1 (IM2[iKD]—shIM1) were grown for 7 days in the absence or presence of doxycycline (2 μg/ml) (− dox or + dox, respectively). mRNAs of IMPDH1 and IMPDH2 were quantified by real-time qPCR in 2 independent experiments carried out in triplicate and normalized to β-actin mRNA (C). Whole cell extracts were analyzed on Western blots with the indicated antibodies. GAPDH was used as a loading control. Quantification of band intensity of p53 and p21 is shown. Mean ± SD is representative of four independent experiments. (right panel) (D).

E Western blots in HCT116 cells treated for 24 h with the indicated concentration of MPA, in the presence of dimethyl sulfoxide (DMSO) or 400 μM guanosine. GAPDH was used as a loading control.

Data information: In panels (B–D), data are presented as relative to control. *P < 0.05, **P < 0.01, ***P < 0.001 by two-tailed Student's t-test.

## Severe nucleotide imbalance leads to increased p21 protein degradation

The levels of p21, an important marker of IRBC activation, fall at higher concentrations of IMPDH inhibitors (Fig 1A, and Appendix Fig S1A and E). We hypothesized that decreased guanine nucleotide synthesis would inhibit transcription of nascent 5S rRNA, IRBC complex formation, and *p21* transcription. Unexpectedly, the results show that p21 mRNA levels are similarly increased after treatment with 1 μM or 10 μM MPA, though to a lower extent than induced by ActD treatment (Fig 3A), suggesting that the transcriptional activity of p53 toward p21 is not differently affected by increasing concentrations of MPA. Likewise, Pol II transcription did not appear to be largely affected, by analyzing a number of mRNA transcripts (Appendix Fig S3A) nor in using a Renilla luciferase reporter (Appendix Fig S3B). However, increasing concentrations of MPA led to a strong dose-dependent decrease in Pol I transcribed 47S rRNA, as measured by the analysis of the Internal Transcribed Spacers (ITS) 1 and ITS2 (Appendix Fig S3A), and to a reduction in mature 18S and 28S rRNA, measured by [3]H-uridine pulse-chase labeling of nascent rRNA (Fig 3B). These findings are consistent with the induced nucleolar disruption by MPA, evidenced by the redistribution of upstream binding factor (UBF) and fibrillarin to adjacent nucleolar cap structures (Appendix Fig S3C), similar to our earlier findings (Fumagalli et al, 2012) and those of others (Shav-Tal et al, 2005). The inhibition of ribosome biogenesis also agreed with a 30% decrease in [3]H-leucine incorporation into nascent proteins, which is further reduced at 48 h (Appendix Fig S3D). In parallel to inhibition of Pol I transcription, 10 μM MPA treatment led to a strong inhibition of 5S rRNA, and 5.8S rRNA synthesis, as compared to

1 μM MPA and ActD (Fig 3C). However, there was no change in the amount of RPL5 and RPL11 co-immunoprecipitating with HDM2, in cell extracts cleared of mature ribosomes, after treatment with 1 versus 10 μM MPA (Fig 3D), though there was a small reduction in the amount of associated 5S rRNA (Fig 3E). Consistent with p21 mRNA levels (Fig 3A), we found higher levels of IRBC complex components bound to HDM2 in the presence of ActD (Fig 3D and E). Similar results were obtained with the immunoprecipitation of endogenous RPL5 (Appendix Fig S3E and F). Thus, the reduction in 5S rRNA synthesis by GMP depletion did not appear to limit formation of the IRBC complex, supporting the finding that p21 transcription is not differentially regulated as a function of the dose of MPA.

The results suggested that the effects on p21 were regulated at the translational or post-transcriptional level. The half-life of p21 is known to be very short (Abbas & Dutta, 2009), raising the possibility that the decrease in p21 protein levels is attributed to a decrease in its half-life. To test this possibility, cells were pre-incubated with 1 μM or 10 μM MPA for 24 h, then treated with cycloheximide for increasing times. The results show that the half-life of p21 is dramatically reduced at 10 μM MPA versus 1 μM MPA, whereas that of p53 is enhanced, showing differential regulation of p21 versus p53 following guanine nucleotide shortage (Fig 3F). Moreover, p21 degradation appears to be due to enhanced proteasomal degradation, as MG132 had little effect in cells pretreated with 1 μM MPA, but dramatically enhanced p21 levels in cells treated with 10 μM MPA (Fig 3G), whereas the opposite effect was observed for p53 protein levels (Appendix Fig S3G). Taken together, the results argue that the decrease in p21 levels by severe IMPDH inhibition is due to its enhanced proteasomal degradation.

**Figure 2.  The IRBC and ATR distinctly regulate p53 stabilization upon nucleotide depletion.**

A HCT116 cells were transfected with either a NT or a siRNA against RPL11 for 24 h and treated with the vehicle alone (−) or the indicated concentration of MPA for 24 h. The levels of p53 and p21 were analyzed on Western blots. GAPDH was used as a loading control. Quantification of band intensity of p53 and p21 of four independent experiments is shown (right panel).

B HCT116 cells were transfected with the indicated siRNA and 24 h later treated with the vehicle alone or 10 μM MPA for 6 h, and the levels of p53 and p21 were analyzed on Western blots. GAPDH was used as a loading control. Quantification of band intensity of p53 of at least four independent experiments is shown (right panel).

C HCT116 cells were transfected with either a NT siRNA or a siRNA against RPL11 for 24 h and were preincubated for 30 min in the absence (−) or the presence (+) of the combination of 10 μM of the ATR inhibitor VE-821 (ATRi) and 10 μM the ATM inhibitor KU-55933 (ATMi), followed by addition of the vehicle alone (−) or MPA 10 μM for additional 24 h. The levels of p53 were analyzed on Western blots. GAPDH was used as a loading control. Quantification of band intensity of p53 of at least two independent experiments is shown (right panel).

Data information: All data are presented as Mean ± SD, relative to control. *P < 0.05, **P < 0.01, ***P < 0.001, by two-tailed Student's t-test.

▶

**A**

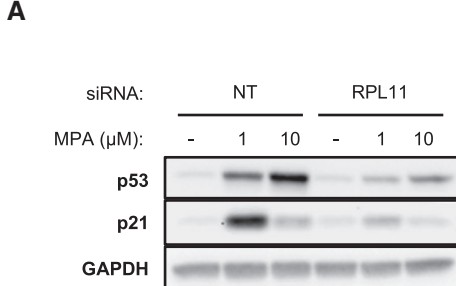

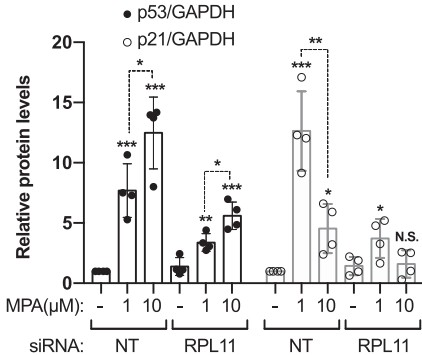

**B**

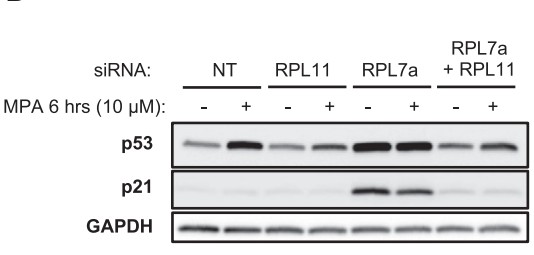

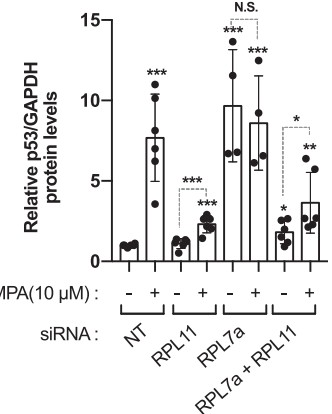

**C**

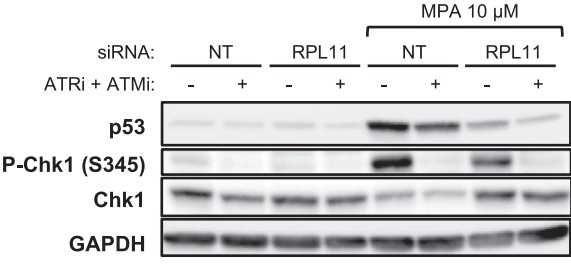

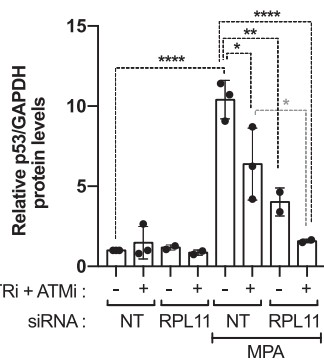

Figure 2.

## Severe nucleotide imbalance overcomes the IRBC-induced G1/S checkpoint leading to replicative stress

The downregulation of p21 and the induction of replicative stress by higher doses of MPA (Fig 1A) indicate that cells bypass the G1/S checkpoint upon severe nucleotide depletion. To test this possibility, we first analyzed cell-cycle profiles of cells treated with either MPA or ActD for 24 h. Unlike cells treated with 1 μM MPA, largely arrested in

G0/G1 and G2/M, recapitulating the IRBC-dependent effects of ActD-mediated Pol I inhibition, a large proportion of cells treated with 10 μM MPA were found in early S phase (Fig 4A and Appendix Fig S4A; Fumagalli *et al*, 2012). Importantly, this event appears to be a direct consequence of a compromised G1/S checkpoint, as cells first synchronized in G1 by serum deprivation and re-stimulated with serum in the presence of 10 μM MPA, also initially accumulate in early S phase (Appendix Fig S4B) and appear to slowly progress

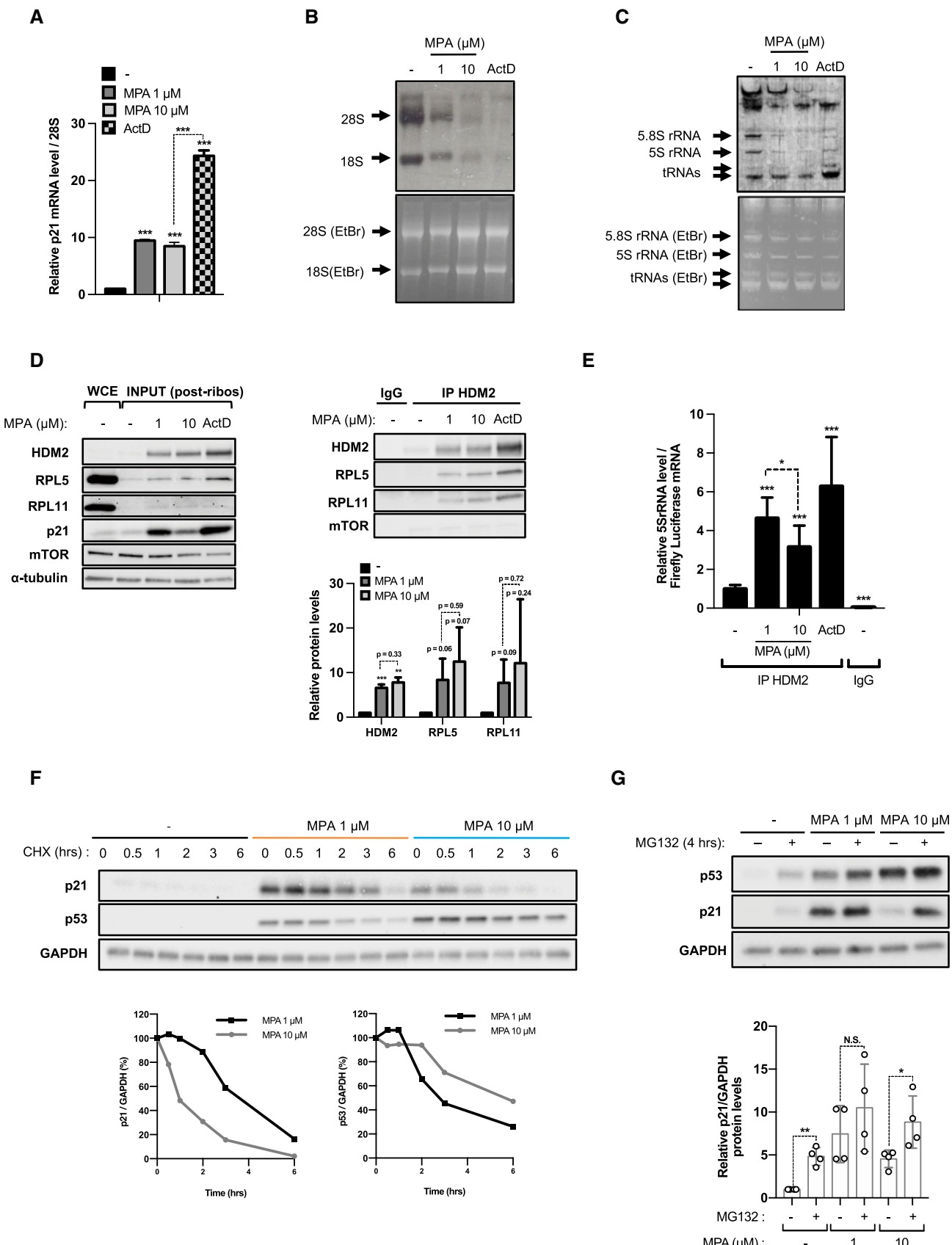

**Figure 3.**

◀

**Figure 3.  Protein stability of p21, but not formation of the IRBC complex, is reduced by higher concentrations of MPA.**

A    HCT116 cells were treated with either vehicle alone (−), 1 μM MPA, 10 μM MPA, or 5 nM ActD for 24 h and mRNA levels of p21 were quantified by real-time qPCR in 2 independent experiments carried out in triplicate and normalized to 28S rRNA.

B    Autoradiogram (upper panel) and Ethidium Bromide (EtBr)-stained agarose gel (lower panel) of 18S and 28S rRNA from HCT116 cells treated as indicated in the Materials and Methods section.

C    Autoradiogram (upper panel) and EtBr-stained TBE-urea polyacrylamide gel (lower panel) of 5S, 5.8S rRNA and tRNAs in lysates from (B).

D, E  HCT116 cells were treated as in (B) for 24 h. Cell lysates were collected, subjected to ultracentrifugation, spiked with firefly luciferase mRNA, and immunoprecipitated with anti-HDM2 rabbit antibody (IP HDM2) or the IgG control (IgG). (D) Levels of HDM2, RPL5, RPL11, and the IP-negative control mTOR in the whole cell extract (WCE), the post-ribosomal lysates (INPUT, left panel) and in the HDM2 immunoprecipitated fraction (right panel) were analyzed on Western blots. The results are representative of two independent experiments. Quantification of band intensity of HDM2, RPL5, and RPL11 of at least two independent experiments is shown (lower right panel). (E) Levels of immunoprecipitated 5S rRNA were determined by real-time qPCR and normalized to the firefly luciferase mRNA. Data are normalized and presented as relative to untreated cells immunoprecipitated with anti-HDM2. The results are representative of three independent experiments carried out in triplicates.

F    HCT116 cells were pretreated with vehicle alone (−), 1 μM, or 10 μM MPA for 24 h, and the levels of p53 and p21 were followed on Western blots in presence of 100 μg/ml cycloheximide (CHX) for the indicated time, in the continued presence of the treatment. GAPDH was used as a loading control. Quantification of band intensities is shown in the lower panels as relative to time 0 of CHX treatment.

G     HCT116 cells were pretreated as in (F), and 10 μM of the proteasome inhibitor MG132 was added for 4 h when indicated. The levels of p53 and p21 were analyzed on Western blots. GAPDH was used as a loading control. Quantification of p21 band intensity of at least four independent experiments is shown in the lower panel.

Data information: In panels (A, E, and G), data are presented as mean ± SD, relative to control. *$P < 0.05$, **$P < 0.01$, ***$P < 0.001$, by two-tailed Student's $t$-test.

through the cell cycle (Appendix Fig S4C). The effects of MPA on cell-cycle progression were not ascribed to cellular context as similar results were observed in a number of CRC cell lines, including RKO, LoVo, and LS174, harboring oncogenic alterations in K-Ras, and/or B-Raf, and mutations in the APC-Wnt-ß-catenin pathway leading to c-Myc upregulation (TCGA, 2012), but all having a functional p53 (Fig 4B and Appendix Fig S4D). As importantly, the analysis of p53, p21, and Chk1 S345 phosphorylation levels suggests that the differential activation of the IRBC and the induction of replicative stress as a function of MPA concentrations is a conserved molecular response in CRC cell lines (Appendix Fig S4E).

Consistent with cell-cycle profiles (Fig 4A), the analyses of a number of proteins involved in regulating cell-cycle progression show that treatment with 1 μM MPA or ActD decreases the levels of cyclin A and the phosphorylation of retinoblastoma (RB) S780, whereas levels of p21 and cyclin D1 increase (Appendix Fig S4F). In contrast, 10 μM MPA-treated cells display reduced levels of p21, and cyclin D1 similar to untreated cells, while the levels of RB S780 phosphorylation and cyclin A, required for S phase progression, are increased, compatible with a compromised G1/S checkpoint (Appendix Fig S4F). Moreover, these cells are delayed in completing S phase (Appendix Fig S4C), indicating a hindrance in DNA replication, potentially ascribed to the reduced levels of dGTP (Fig 1B), a common cause of replication fork stalling (Awasthi et al, 2015). To test this hypothesis, we monitored the speed of DNA replication by single-molecule DNA fiber analysis, employing two nucleotide analogues (Ercilla et al, 2016; Appendix Fig S4G), showing that MPA-treated cells display a greatly reduced rate of DNA fiber elongation as compared to non-treated cells (Fig 4C and D). Importantly, reduced rate of DNA fiber elongation was associated with asymmetric replication of the two DNA strands indicative of replicative stress (Fig 4C, right insert). We also detected an increase in ssDNA generation, by BrdU incorporation under non-denaturing conditions and accumulation of chromatin-loaded RPA, in cells treated with MPA, compared with hydroxyurea (HU) which causes rapid depletion of dNTPs (Fig 4E and Appendix Fig S4H). This result suggests that MPA leads to replicative fork arrest, and to the uncoupling between replicative helicase and DNA polymerases. Moreover, after prolonged nucleotide imbalance, Chk1 is downregulated and the

levels of ATM-dependent phosphorylation of H2AX S139 and Chk2 T68 increase (Fig 4F). This finding, combined with H2AX phosphorylation occurring exclusively in DNA replicating cells (Appendix Fig S4I and J), and the enhancement of γH2AX by ATR inhibition in MPA-treated cells (Appendix Fig S2C) indicate that prolonged nucleotide imbalance leads to persistent fork stalling and the generation of DSBs, likely due to the collapse of the replicative forks (Saintigny et al, 2001; Zhang et al, 2005; Hanada et al, 2006; Toledo et al, 2013). Moreover, the results further suggest that severe nucleotide depletion impedes G1-arrest, allowing cells to enter S phase where they encounter replicative stress, which with time leads to DNA damage.

### The suppression of p21 drives S phase entry upon nucleotide imbalance

To determine whether the loss of a robust p21 response is involved in the failure of the IRBC to preserve cells in G1, we first analyzed the effects of MPA on the cell-cycle progression, as a function of p21 protein levels. We initially set a threshold for p21 analyses based on antibody detection of the protein in isogenic HCT116 $p21^{-/-}$ cells and discriminated two cellular populations in the parental cells, p21 low (p21$^{low}$) and p21 high (p21$^{high}$). Cell-cycle analyses of total population (Fig 5A, top panel) show similar profiles as described above (Fig 4A and Appendix Fig S4A). In the absence of treatment, p21$^{high}$ cells represented 11.4% of the total population, but this proportion reaches 85.6% of total cells in the presence of 1 μM MPA, and only 38.0% when treated with 10 μM MPA (Fig 5A). Critically, p21$^{high}$ cells are mostly arrested in G0/G1 and the G2/M regardless of MPA concentration, whereas p21$^{low}$ cells treated with 10 μM MPA are mostly found in early S phase. A parallel comparative analysis of HCT116 wild-type and $p21^{-/-}$ isogenic cells showed that 10 μM MPA strongly reduces the proportion of $p21^{-/-}$ cells present in G1, while enriching the proportion of cells in early S phase (Appendix Fig S5A), consistent with a key role of p21 in mediating the effects of MPA on cell-cycle regulation, downstream of the IRBC and p53.

To confirm that a reduction in p21 levels is a key determinant of S phase entry upon nucleotide depletion, an empty plasmid (EV) or a plasmid encoding a p21$^{wt}$ cDNA under the control of an ectopic

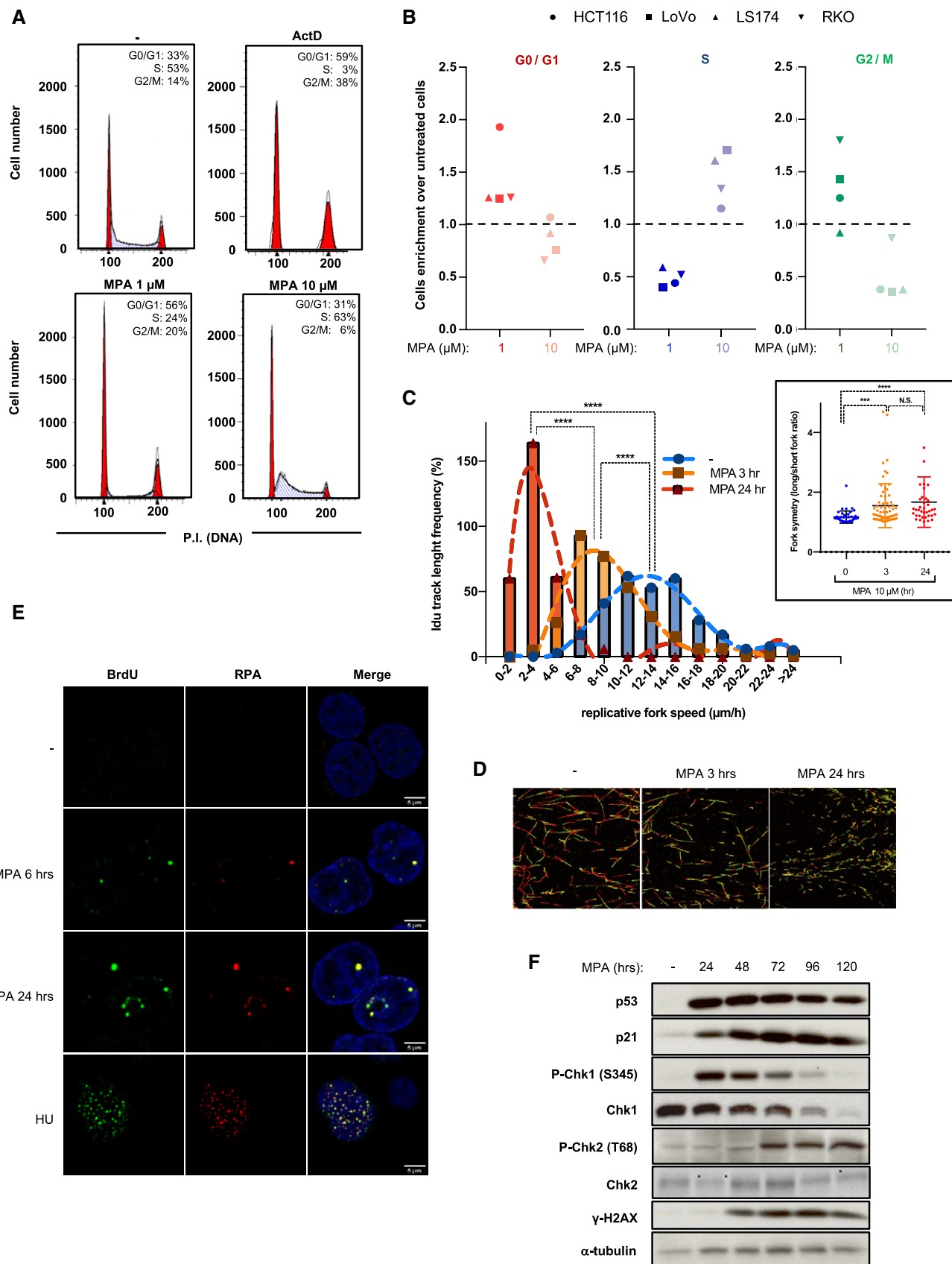

**Figure 4.**

**Figure 4. S phase entry and inhibition of DNA replication are induced at high concentrations of MPA.**

A  HCT116 were treated as indicated for 24 h, subjected to propidium iodide staining, and analyzed by fluorescence-activated cell sorting (FACS). Data are representative of at least three independent experiments.

B  Colon adenocarcinoma cell lines HCT116 (circles), LoVo (squares), LS174 (triangles), or RKO (inversed triangles) were treated with the vehicle alone (−), 1 μM MPA or 10 μM MPA for 24 h and subjected to propidium iodide staining and FACS analysis. Data are shown as the fold induction of the cell number in G0/G1 phases (blue), S phase (red), and G2/M phases upon treatment with 1 μM MPA (dark) or 10 μM MPA (clear) over vehicle-treated cells.

C  IdU track length distribution, and distribution of long fork/short fork IdU track length ratios (insert). IdU track length of at least 300 fibers was measured for each condition. For fork symmetry analyses, IdU track length was measure in at least 38 bidirectional forks (see Materials and Methods). ***$P < 0.001$, ****$P < 0.0001$ by Kruskal–Wallis test of the means.

D  Representative images from (C).

E  HCT116 cells, pre-incubated with 10 μM BrdU for 48 h, were treated with Vehicle (−), 10 μM MPA for 6 or 24 h, or 20 μM Hydroxyurea for 4 h before cell permeabilization and fixation (see Materials and Methods). Cells were immunostained for native BrdU (green) or RPA (red). DNA was counterstained with DAPI (blue). One representative z-confocal stack is shown per condition. Scale bars correspond to 5 μm.

F  HCT116 cells treated with 10 μM MPA for increasing time were analyzed by Western blots for the indicated proteins. α-tubulin was used as a loading control.

promoter (p21-OE) was transiently introduced into $p21^{-/-}$ cells (Appendix Fig S5B) and cells were synchronized in G1 and serum stimulated as described above (Appendix Fig S4B). Analysis of the cell-cycle distribution of p21-OE cells was analyzed in $p21^{high}$ versus $p21^{low}$ populations and showed that the $p21^{high}$ cells were unable to bypass the G1/S checkpoint, even in the presence of 10 μM MPA (Fig 5B). These results support that at higher doses of MPA the induction of p21 is hindered, allowing cells to progress into S phase.

**The IRBC protects from replicative stress and DNA damage upon nucleotide imbalance**

As the IRBC complex triggers *p21* transcription, and its loss reduces MPA-induced p21 expression (Fig 2A), we predicted that the inhibition of IRBC complex formation would further enhance the ability of MPA to drive cells into S phase. Thus, cells were depleted of either RPL11 or p53, prior to G1 synchronization and serum stimulation, as above. In either condition, serum deprivation leads to the same extent of G1 arrest (Fig 6A and Appendix Fig S6A). In untreated cells, p53 depletion does not greatly alter the cell cycle, whereas RPL11 depletion led to an increase in the proportion of cells in S phase, likely due to the slower progression of RPL11-deficient cells through the cell cycle, as we previously reported (Teng *et al*, 2013; Fig 6A and Appendix Fig S6A). Importantly, the depletion of RPL11 or p53 potentiates MPA-induced accumulation of cells in S phase, particularly in early S phase (Fig 6A, and Appendix Fig S6A and B). As the downregulation of the IRBC allows DNA replication, despite low nucleotide availability, this could favor the occurrence of replicative fork stalling and DNA damage. Unexpectedly, after 72 h, we found by Western blot analysis that not only MPA, but also the depletion of RPL11 alone induced γ-H2AX (Fig 6B and Appendix Fig S6C). Moreover, RPL11 depletion further increased the effect of MPA on γ-H2AX, while suppressing the induction of p53 (Fig 6B). These results were confirmed by quantitative microscopy-based single-cell analyses of γ-H2AX or 53BP1, both indicative of DNA-double strand breaks and the activation of the DDR machinery (Fig 6C–E and Appendix Fig S6C). Similar results were obtained with the alkaline comet assay, indicating global DNA damage, predominantly single-strand nicks and gaps (Fig 6F and Appendix Fig S6C). To assess whether this response was due to the loss of the IRBC or to the impairment of ribosome biogenesis, we depleted cells of RPL5 or RPS6 (or eS6). While the depletion of RPL5 induced γ-H2AX and amplified MPA-induced response, the depletion of RPS6 had no such effect, but reduced the effect of MPA (Appendix Fig S6C–H).

Likewise if the synthesis of 5S rRNA is first blocked by depleting cells of TFIIIa, required for Pol III-mediated 5S rRNA transcription (Appendix Fig S6I and J), we also observed the induction of γ-H2AX and the amplification of the response in cells treated with 10 μM MPA, as well as a reduction in p53 induction by MPA (Appendix Fig S6K–M). These results show that unexpectedly the inhibition of the IRBC alone induces DNA damage. Moreover, it amplifies those caused by IMPDH inhibition, potentially due to the favored S phase entry despite nucleotide levels shortage leading to replicative stress and DNA damage (see Discussion and model Fig 7), suggesting that the IRBC serves as barrier against genomic instability caused by limiting pools of GMP.

# Discussion

This study demonstrates a hierarchy of p53-dependent checkpoints activated by the inhibition of *de novo* guanine nucleotide synthesis. We have focused on IMPDH inhibitors given their clinical approval in other disease settings and the recent exciting pre-clinical findings concerning their application in specific cancer types (Valvezan *et al*, 2017; Huang *et al*, 2018); however, it is most likely that other clinically approved non-genotoxic nucleotide inhibitors operate in a similar manner. We show that the primary response to IMPDH inhibition is the induction of the IRBC and p21-mediated G1 arrest, consistent with ribosome biogenesis being the most nucleotide demanding metabolic process in proliferating cells (Pelletier *et al*, 2018). However, if guanine nucleotide pools continue to fall, the levels of p21 protein are diminished, allowing cells to enter S phase, where they encounter replicative stress and DNA damage, an effect that is strongly enhanced by disruption of the IRBC, and which is known to be a major cause of genomic instability (Gaillard *et al*, 2015). Interestingly, a crosstalk between the DDR and ribosome biogenesis has recently been described, where activation of ATM leads to Treacle recruiting Nijmegen Breakage Syndrome protein 1 (NBS1) to the nucleolus, where this complex suppresses rDNA transcription (Ciccia *et al*, 2014; Larsen *et al*, 2014). This regulatory mechanism is an important player in the conservation of the stability of the highly repetitive and actively transcribed rDNA genomic regions (Pelletier *et al*, 2018). The ATM/NBS1/Treacle response is particularly critical during rDNA replication, where such conflicts lead to the formation of R-loops, rRNA:DNA hybrids, giving rise to DNA-double strand breaks (Pelletier *et al*, 2018). Thus, the importance of ribosome biogenesis in replicative stress is underscored by

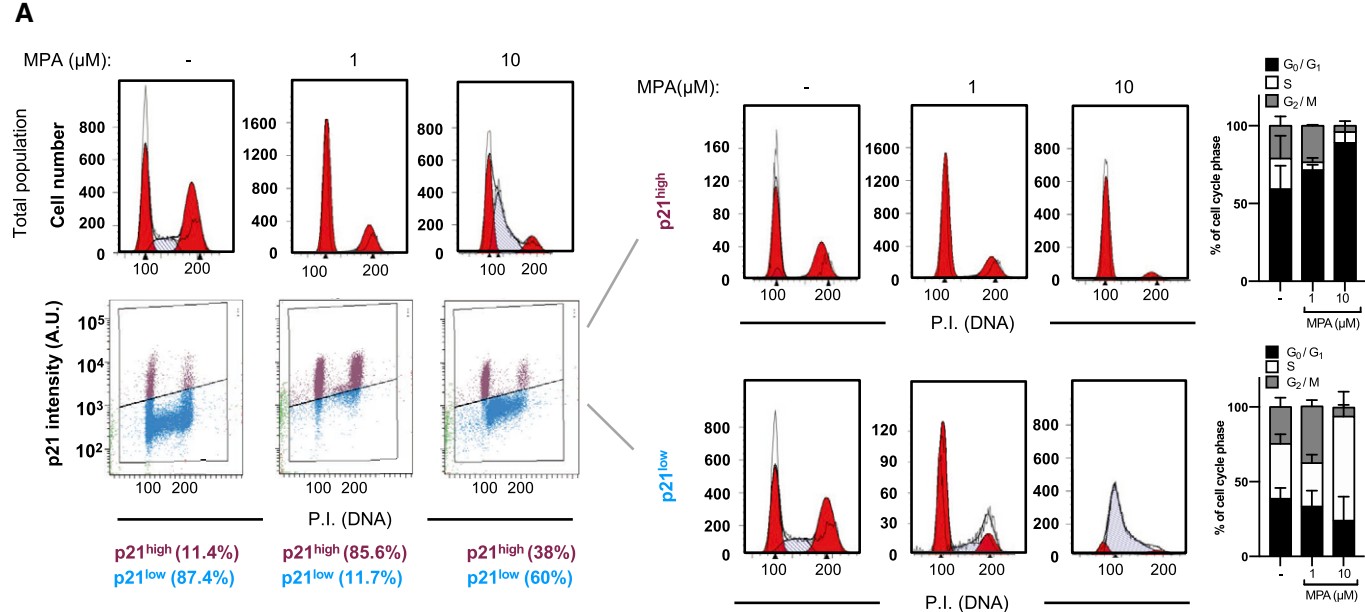

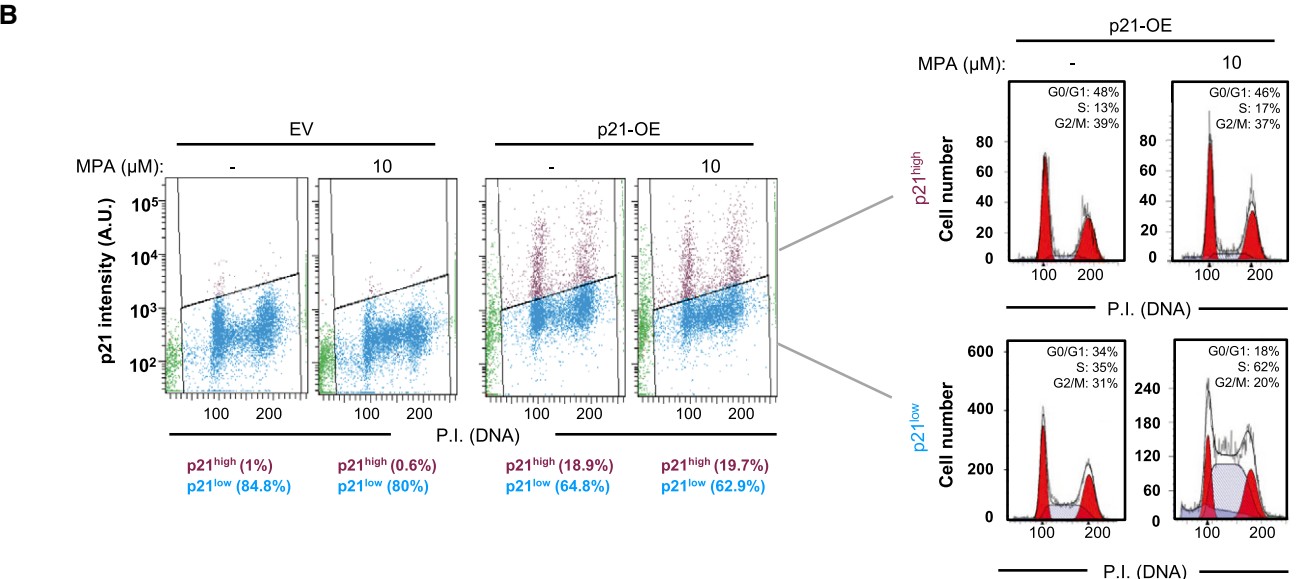

**Figure 5. The downregulation of p21 facilitates S phase entry at high concentrations of MPA.**

A HCT116 cells were treated with either the vehicle alone (−), 1 μM MPA, or 10 μM MPA for 24 h, before p21 immunolabeling, propidium iodide staining, and FACS analysis. Cell-cycle profiles of the total cell population (upper panel, left), the total intensity of p21, and propidium iodide were determined in 20,000 cells and plotted in a scatter diagram (lower panel, left). The specific cell-cycle profiles are shown of p21$^{high}$ (upper panel, right) and p21$^{low}$ (lower panel, right) cell populations. Mean ± SD of the percentage of cells in each cell-cycle phase of at least two independent experiments is shown in the right panels.

B HCT116 p21$^{−/−}$ isogenic cell lines were transfected for 24 h with an empty plasmid (EV) or a plasmid encoding a p21$^{wt}$ cDNA (p21-OE), and then, cells were serum-deprived for 16 h before the re-addition of serum for additional 24 h in the absence or presence of 10 μM MPA. Cells were subjected to p21 immunolabeling, propidium iodide staining, and FACS analysis. The results are representative of two independent experiments.

both the IRBC-p21 response and the NBS1/Treacle signaling pathway, each acting at a distinct level to protect from DNA damage.

The loss of p21 with increasing severity of IMPDH inhibition seems responsible for the entry of cells in S phase, as ectopic expression of *p21* in MPA-treated HCT116 *p21*$^{−/−}$ cells is sufficient to maintain cells in G1. The downregulation of p21 appears to be independent of its transcription, but caused instead by its selective degradation. It has been recently reported in a context of doxorubicin-

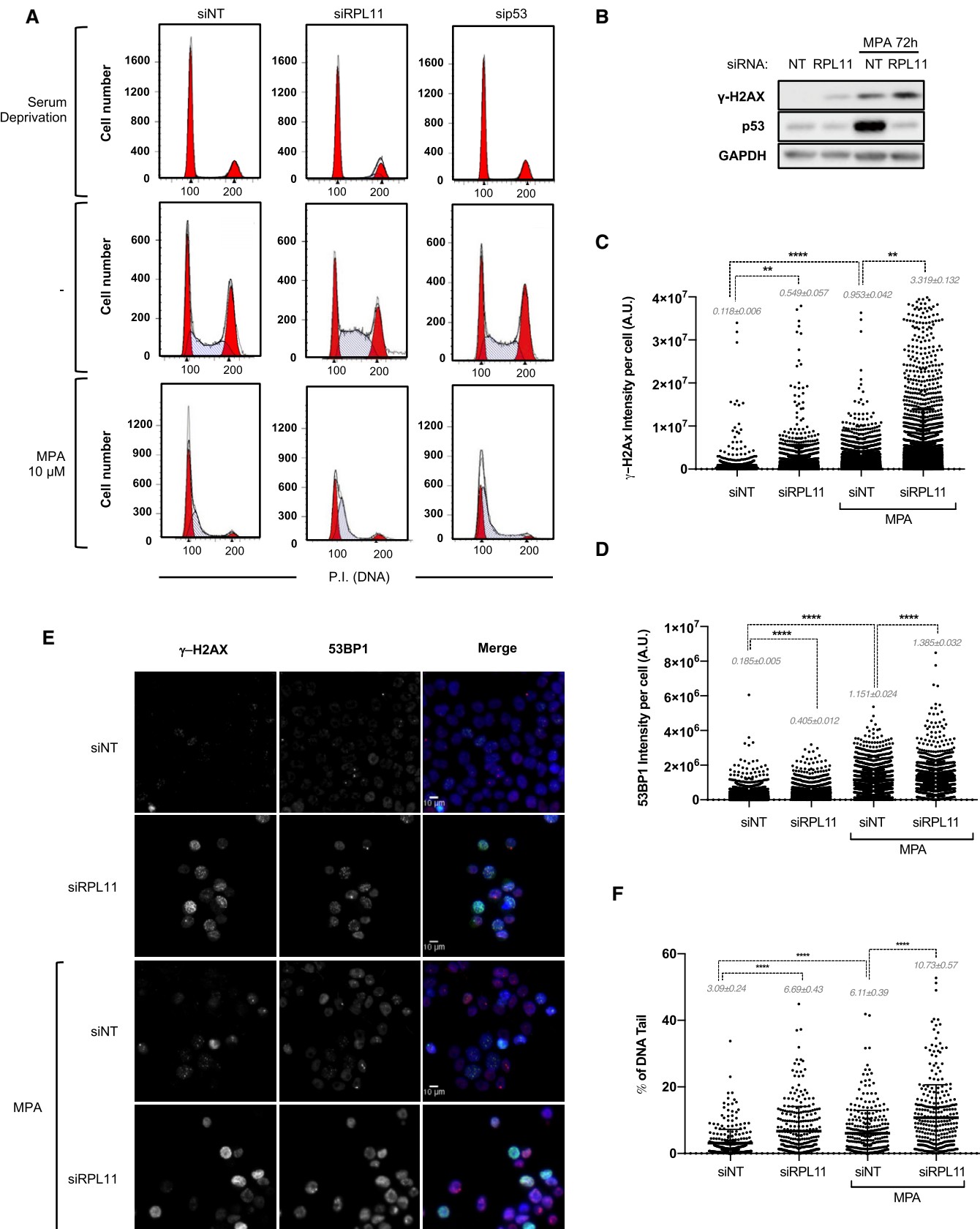

**Figure 6.**

**Figure 6. Disruption of the IRBC enhances MPA-mediated DNA damage.**

A  HCT116 cells were transfected for 8 h with the indicated siRNA then treated and analyzed as in Fig 5B. Here are shown the cell-cycle profiles after 16 h of serum deprivation (top panels), or after an additional 24 h of serum re-addition in the absence (middle panels), or presence of 10 μM MPA (bottom panels). The results are representative of at least three independent experiments.

B  HCT116 cells transfected with a NT siRNA or a siRNA against RPL11 for 24 h were treated with 10 μM MPA for additional 72 h. The levels of γ-H2AX were analyzed on Western blots. GAPDH was used as a loading control.

C, D  Statistical analysis of the immunostaining of γ-H2AX (C) or 53BP1 (D) in HCT116 cells treated as in (B) (see Materials and Methods). Total nuclear intensities were obtained on > 6,000 (γ-H2AX) and > 1,000 (53BP1) individual cells for each condition from four or two independent biological replicates, respectively. **$P < 0.01$, ****$P < 0.0001$ by Kruskal–Wallis test. A.U. Arbitrary Units.

E  Representative images from (C, D) of cells labeled with γ-H2AX (green), 53BP1 (red), and DAPI (blue). Scale bars correspond to 10 μm.

F  HCT116 cells treated as in (B) were analyzed by the comet assay. The percentage of tail DNA/total DNA was analyzed in a total of 300 cells for each condition from three independent biological replicates. ****$P < 0.0001$ by Kruskal–Wallis test.

induced DNA double-strand breaks that the proteasomal degradation of p21 during S phase is prevented by Chk1 inhibition, suggesting a role for Chk1 signaling in p21 degradation (Hsu *et al*, 2019). However in the case of IMPDH inhibitors, replicative stress and Chk1 activation would require S phase entry, restricted by p21. In normal cycling cells, the degradation of p21 takes place prior to the G1/S transition by the ubiquitin ligases CRL4^Cdt2 and SCF^Skp2 suggesting a potential role of these proteins in the degradation of p21 by IMPDH inhibitors (Barr *et al*, 2017; Sheng *et al*, 2019). However, it has also been reported that p21 binds to the central domain of HDM2 (Jin *et al*, 2003; Zhang *et al*, 2004), where RPL11 also binds (Zheng *et al*, 2015), and that HDM2 promotes the degradation of p21 in tumor cells which are either deficient or mutated in p53 (Jin *et al*, 2003; Zhang *et al*, 2004). Given that higher doses of MPA lead to p53 S15 phosphorylation, which is reported to disrupt the HDM2/p53 interaction, this would liberate HDM2 to degrade p21. It should be noted that the effects of HDM2 on p21 are independent of the HDM2 E3 ligase function, probably acting through the binding of both p21 and the C8 subunit of the 20S proteasome (Jin *et al*, 2003; Zhang *et al*, 2004). It will be of interest to determine the underlying mechanism(s) by which severe IMPDH inhibition leads to the degradation of p21 protein.

During early tumorigenesis, oncogenic stresses lead to accelerated rates of rRNA synthesis despite nucleotide deficiencies (Bester *et al*, 2011), which may impair ribosome biogenesis, and lead to the IRBC-dependent senescence phenotype (Nishimura *et al*, 2015). The acute sensitivity of ribosome biogenesis impairment to nucleotide depletion as compared to that of replicative stress supports the notion that the IRBC acts as a first-line barrier against oncogenes. Consistent with this model, earlier studies conducted in *Eμ-Myc* transgenic mice harboring the C305F mutation in MDM2, which disrupts its interaction with RPL11 and the inhibitory effect of the IRBC complex, demonstrated that mutant mice succumb earlier to lymphoma with respect to their WT counterparts (Macias *et al*, 2010). As the loss of the IRBC enhances MPA-induced DNA damage in cell culture, one might expect the MDM2-mutant tumors, which are IRBC-defective, to have an increased rate of genomic instability. The mechanism by which the disruption of the IRBC leads to DNA damage is not known. An interpretation for the amplification of DNA damage may come from studies showing that in RPL11 deleted mouse embryo fibroblasts, the expression of genes involved in DNA repair are downregulated (Morgado-Palacin *et al*, 2015). On the other hand, cells harboring reductions or mutations in components of the IRBC continue to cycle, but with reduced ribosome content and translational capacity, leading to alterations in the pattern of

translation, as we reported earlier (Teng *et al*, 2013). It is likely that under such conditions the translation of specific factors is required to sustain the correct replication of DNA, or to sense and resolve replicative stress to limit DNA damage and potentially genomic instability.

Activation of the IRBC would appear to have an important role in cancer therapy. Many approved chemotherapeutic agents are known to interfere with ribosome biogenesis (Burger *et al*, 2010). It has been long established that the major effects of 5-FU on p53-induced apoptosis are not due to the inhibition of thymidylate synthase, but to the inhibition of rRNA synthesis (Pritchard *et al*, 1997; Sun *et al*, 2007). Likewise recent studies showed that oxaliplatin does not induce p53 by causing DNA damage, but instead through impairing ribosome biogenesis and the apparent activation of the IRBC (Bruno *et al*, 2017). These insights have led to the development of new drugs specifically designed to attack ribosome biogenesis, particularly CX-5461 (Drygin *et al*, 2011; Bywater *et al*, 2012), with the first results published of a phase 1 dose escalation clinical trial in patients with advanced hematologic cancers (Khot *et al*, 2019). Importantly, a recent report showed that the IMPDH inhibitor mizoribine, selectively targets tumors developing in TSC2-deficient mice, which display constitutive mTORC1 activation (Valvezan *et al*, 2017). mTORC1 coordinates the upregulation of ribosome biogenesis by increasing Pol I- and Pol III-dependent transcription, and controlling the translation of RP mRNAs (Pelletier *et al*, 2018). In this setting, nucleotide pools are apparently diverted to rRNA synthesis to sustain the hyperactivation of ribosome biogenesis, which appears to sensitize TSC2-deficient cells to replicative stress caused by guanine nucleotide depletion. Although Valvezan *et al* show that IMPDH inhibition led to increased Chk1 phosphorylation and DNA damage in TSC2-deficient models, there was no apparent effect on rRNA synthesis in either *TSC2^{-/-}* or *TSC2^{+/+}* cells (Valvezan *et al*, 2017). It is difficult to rationalize this observation with earlier reports (Huang *et al*, 2008; Sun *et al*, 2008), the recent findings of Huang *et al*, in c-Myc overexpressing SCLCs (Huang *et al*, 2018), and those presented here, particularly with the high demand of nucleotides in ribosome biogenesis versus DNA synthesis. Given the potential importance in developing inhibitors of ribosome biogenesis to treat aggressive cancers, such as those driven by c-Myc, it will be critical to understand whether TSC is unique in its response to IMPDH inhibitors or whether it can be more broadly applied to other cancer types. Moreover, given the role of the IRBC as a tumor suppressor, a deeper knowledge of the biological processes controlling the hierarchy of intrinsic tumor checkpoints will be

critical for the development of novel strategies exploiting anti-cancer agents in the clinic.

# Materials and Methods

### Cell culture

HCT116, LoVo, LS174, and RKO human colorectal carcinoma cell lines were obtained from the American Type Culture Collection and maintained in Dulbecco's modified Eagle medium (DMEM) supplemented with 10% heat-inactivated fetal bovine serum (Sigma-Aldrich, St Louis, MO, USA). Isogenic HCT116 cells disrupted for p53 or p21 expression were kindly provided by Dr. Vogelstein (Johns Hopkins University of Medicine, Baltimore, MD, USA).

### Stable cell lines generation

The shRNA sequences for IMPDH1 and IMPDH2 were selected using the Designer of Small Interfering RNA (http://biodev.cea.fr/DSIR/DSIR.html). shRNA targeting IMPDH2 were cloned into the retroviral TRMPVIR vector as previously described (Zuber *et al*, 2011). Five shRNA sequences were screened, and two were chosen for virus production (see Appendix Table S2). The generation of shRNA encoding retrovirus particles and the transduction of HCT116 cells were performed as previously described (Zuber *et al*, 2011). After 24 h of transduction, the cells were sorted for Venus-positive cells by FACS (MoFlo Astrios cell sorter). Individual clones were grown, treated with doxycycline (2 μg/ml), and analyzed for mRNA and protein expression. Separately, control non-targeting (shNT) or a shRNA against IMPDH1 (shIM1) were cloned into pLKO.1 plasmid and lentiviral particles were generated as described previously (Fonseca *et al*, 2015; see Appendix Table S2). HCT116 cells stably expressing IMPDH2 shRNA were transduced with control shRNA or IMPDH1 shRNA followed by 3 weeks of puromycin selection leading to the generation of IM2$^{iKD\text{-}shNT}$ or IM2$^{iKD\text{-}shIM1}$, respectively. HCT116 cells stably expressing IMPDH2 shRNA seq#2 (see Appendix Table S2) were transduced with control shRNA leading to the generation of IM2$^{iKD\#2\text{-}shNT}$.

### Reagents and plasmids

MPA, AVN944, actinomycin D, etoposide, hydroxyurea, KU55933, guanosine, doxycyclin and Cycloheximide, CldU and IdU were purchased from Sigma (Sigma-Aldrich, St Louis, MO, USA). VE-821 (ATRi) was purchased from ApexBio. The transfection reagents, lipofectamine RNAiMAX (for siRNA transfection), lipofectamine 2000 (for plasmid transfection), and TRIzol RNA extraction reagent, were purchased from Invitrogen (Carlsbad, CA, USA). Transfections were performed according to manufacturer's instructions. EN3HANCE autoradiography enhancer, $^3$H-Leucine, and $^3$H-Uridine were purchased from PerkinElmer. The protein assay kit was purchased from Pierce (Rockford, IL, USA). The antibodies used are listed in Appendix Table S1. It should be noted that the anti-IMPDH antibody (pan antibody) was argued to be a specific antibody against IMPDH1, but our findings indicate that it recognizes both IMPDH1 and IMPDH2. The Magna ChIP Protein A/G magnetic beads mix was purchased from Millipore (Billerica, MA, USA). Random

hexamers and MMLV Reverse transcriptase were from Invitrogen (Carlsbad, CA, USA). The SYBR Green kit was purchased from Roche (Basel, Switzerland). The Dual-Luciferase kit was purchased from Promega (Madison, WI, USA). The sequences of siRNAs used in the experiments, siRPL5, siRPL11, siRPL7a, sip53, and siNT are reported in Appendix Table S2. For each treatment, the amount of siRNA transfected was maintained constant between samples, by using siNT where required. pcDNA 3.1 (Addgene plasmid V790-20), PG13-luc plasmids (wt p53 binding sites, Addgene plasmid #16442), and MG15-luc plasmids (mut p53 binding sites, Addgene plasmid #16443) were obtained from Addgene (Cambridge, MA, USA). The pcDNA3.1-p21 plasmid is a kind gift from Dr. Sawaya.

### Protein analysis

Cell protein extracts and protein concentrations for Western blot analysis were carried out with the indicated antibodies (Appendix Table S1) as recently described (Gentilella *et al*, 2017). Immunoblots were developed using secondary horseradish peroxidase-coupled antibodies (Polyclonal swine anti-rabbit and polyclonal rabbit anti-mouse, Agilent, CA, USA) and an enhanced chemiluminescence kit (GE Healthcare). Signal was detected using iBright CL1000 (Thermo Fisher Scientific, PA, USA), and quantification of band intensities by densitometry was carried out using the ImageJ software.

### Real-time PCR

Total cell RNA was isolated using TRIzol reagent (Invitrogen) according to the manufacturer's instructions. Reverse transcription and quantitative real-time PCR were performed as previously described (Gentilella *et al*, 2017). The sequences of primers utilized are reported in Appendix Table S2. For the quantification of RNA normalized to the DNA content, cells were harvested in an equal volume of cold PBS and 50 μl of the cell suspension was lysed in 50 μl of 0.1 N NaOH and DNA was quantified using a NanoDrop 1000 spectrophotometer (Thermo Scientific). The remaining cells were pelleted, lysed in TRIzol, a spike of Firefly luciferase mRNA proportional to the amount of DNA was added, and the samples were processed as described above and the RNA amount normalized to luciferase mRNA.

### Immunoprecipitation

Cells grown in 15-cm dishes were lysed and subjected to immunoprecipitation largely as previously described (Donati *et al*, 2013). After lysis, mature ribosomes were pelleted by ultracentrifugation at 200,000 *g* for 2 h at 4°C and an equivalent amount of protein (1 mg) was incubated at 4°C overnight with rotation with anti-RPL5, anti-HDM2, or anti IgG to a ratio antibody/sample of 4 μg/mg. Magna ChIP Protein A/G magnetic beads (Millipore) were added to the extracts for an additional 2 h at 4°C with rotation and washed following manufacturer's instructions. Beads-containing pellets were resuspended either in protein loading buffer for Western blot analysis or TRIzol reagent, together with a spike of firefly luciferase mRNA (5 ng/mg of precipitated proteins) before RNA purification, to recover immunoprecipitated RNA for 5S rRNA qRT-PCR analysis.

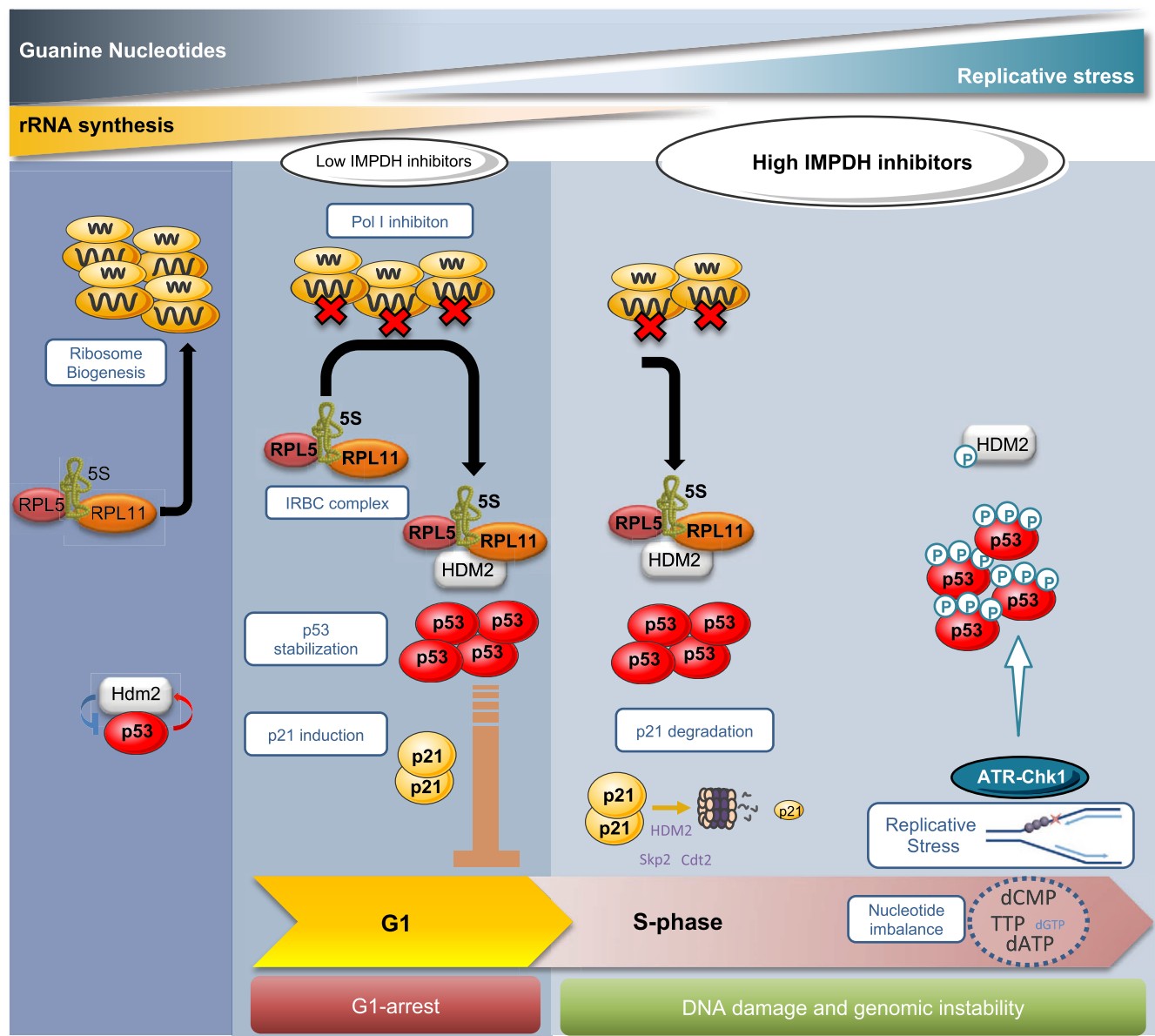

**Figure 7. Model of the hierarchical organization of the IRBC and the DDR response upon ribonucleotides depletion.**

Under normal cell growth conditions, the pre-ribosomal complex formed by RPL5, RPL11, and 5S rRNA is incorporated into nascent ribosomes (Left panel). When nucleotide levels become limiting for 47S rRNA synthesis, ribosome biogenesis is impaired, which converts the RPL5/RPL11/5S rRNA complex into the IRBC complex, which is able to bind and inhibit MDM2, releasing p53, leading to p21 induction and G1 arrest (Middle panel). However, a more severe drop in nucleotide levels leads to the degradation of p21 by the proteasome, impairing IRBC-dependent G1 arrest and allowing S phase entry despite reduced nucleotide pools. Cells are therefore exposed to replicative stress, sensed by the ATR-Chk1 pathway responsible for p53 S15 phosphorylation. Chronic replicative stress leads to DNA damage and eventually genomic instability, key events in tumor progression (Right panel).

**Autoradiographic analysis of rRNA synthesis**

To analyze newly synthesized RNA, cells were pulse-labeled for 2 h with 1.2 μCi/ml of [3H]-uridine (PerkinElmer) and then chased in non-radioactive media for 4 h before TRIzol RNA extraction as described above. 2 μg of total RNA were resolved either on a formaldehyde-containing 1.2% agarose gel for 18S and 28S rRNAs or on a TBE-urea 10% polyacrylamide gel for 5S rRNA, 5.8S rRNA, or tRNAs and transferred to Hybond N+ membrane (GE Healthcare).

After ultraviolet cross-linking, the membranes were sprayed with EN3HANCE (PerkinElmer) and exposed to Kodak BioMax MS film (Kodak) at −80°C for 1 week.

**Measurement of *de novo* protein synthesis by [3H] leucine incorporation**

After treatment, cells were pulse-labeled for 30 min with 10 μCi/ml of [3H] leucine as previously described (Gentilella *et al*, 2017).

Cycloheximide (CHX), at a concentration of 100 μg/ml for 2 h, was used as a control of protein synthesis inhibition.

### Luciferase assay

Cells plated in 6-well plates were transfected with 2 μg of reporter plasmid (PG13-luc or MG-15-luc, see Reagents and Plasmids) and 250 ng of Renilla luciferase control plasmid (Promega, Madison, WI, USA) for 24 h. Cell extracts were prepared and assayed using the Dual Luciferase kit (Promega) according to the manufacturer's instructions. Firefly Luciferase activity was normalized to Renilla luciferase activity.

### Cell-cycle analysis

After treatment, cells were trypsinized, counted, washed, and fixed in cold 70% ethanol solution overnight. For p21 or γ-H2AX immunostaining, cells were permeabilized with 0.1% TritonX-100 for 20 min at room temperature (RT), washed with PBS-Tween20 (0.5%), incubated in 4% FBS–containing PBS-T Blocking Solution for 30 min at RT, and incubated with p21 antibody (1/250, Abcam) or γ-H2AX (Abcam, 1/200) in the blocking solution for 1 h at RT. Cells were then extensively washed in PBS-T, before alexa647-conjugated secondary antibody (Invitrogen, 1/500) incubation for 30 min at RT in the dark. After washing, cells were stained with propidium iodide as previously described (Morcelle *et al*, 2019) before FACS analysis, using FACSCanto II (BD biosciences, CA, USA). In all experiments, 20,000–50,000 gated events were collected. Experiments were analyzed using ModFit LT software (Verity Software House, ME, USA) and BD FACSDiva software (BD Biosciences, CA, USA).

### Immunofluorescence microscopy

Immunofluorescence microscopy was performed largely as previously described (Pelletier *et al*, 2012). Briefly, after 4% formaldehyde fixation for 20 min at RT, and permeabilization with 0.1% Triton X-100 for 10 min at RT, cells were incubated in the blocking solution containing 1% BSA and 0.01% Triton X-100 for 1 h at RT. Cells were then incubated with the indicated primary antibodies (Appendix Table S1) in blocking solution overnight at 4°C. After extensive washes with PBS-T (0.01%), cells were stained with Alexa488- or Alexa555-conjugated secondary antibody (Invitrogen, 1/500 in blocking solution) for 45 min at RT in the dark. After additional washes, cells were mounted with Vectashield mounting solution containing DAPI. Images were generated either with the Leica spectral confocal microscope TCS SP5 with a 63×/1.4 NA objective or with the Zeiss Axio Observer inverted microscope with a 10×/0.25 NA objective. Acquisition settings for the different channels were adjusted and maintained to obtain images in non-saturating conditions. For γ-H2AX and 53BP1 immunostaining quantification, depending on cell confluence, 8–15 images were acquired containing $n > 1,000$ cells per condition. Images were processed and analyzed using ImageJ software (NIH USA) (Toledo *et al*, 2013). The analysis of positive foci for native BrDU and RPA required a pre-extraction protocol. Briefly, after treatment, cells were washed once with PBS and incubated with ice-cold PBS containing Triton X-100 (0.5%) for

1 min on ice prior to 4% formaldehyde fixation. We then proceeded as above using a blocking solution containing 3% BSA and 0.01% Triton X-100.

### DNA fiber assays

DNA fiber assays were largely performed as previously described (Ercilla *et al*, 2016). Cells were treated and pulse-labeled as indicated with CldU (25 μM)/IdU (250 μM). Labeled cells were harvested by trypsinization and resuspended in ice-cold PBS at $5 \times 10^5$ cells/ml. After DNA spreading, fixation in methanol/acetic acid (3:1) solution, denaturation, and neutralization steps, CldU and IdU were detected by immunofluorescence with anti-BrdU monoclonal antibody (Abcam, ab6326; 1/1,000 for CldU labeling and Becton Dickinson, 347580; 1/200 for IdU labeling) for 1.5 h at 37°C. After incubation with primary antibodies, fiber spreads were fixed with 4% PFA-PBS for 10 min at RT and finally incubated with Alexa488- and Alexa555-conjugated secondary antibodies (Invitrogen, 1/500) for 1.5 h at 37°C. Images were obtained using Zeiss LSM880 confocal microscope with a 63× oil immersion objective and then analyzed using Fiji software. The length of at least 300 IdU tracks per condition was measured. We only measured the fibers that were doubly labeled (by CldU and IdU), to exclude from analysis the fibers that were initiated during the second labeling period. To assess fork symmetry, the length of IdU tracks was measured in at least 38 bidirectional forks (labeled by CldU, and IdU in the two directions), and the ratio of length between the longer and the shorter track was analyzed.

### Comet assay

After treatment, cells were harvested by trypsinization, resuspended in ice-cold PBS at $1 \times 10^6$ cells/ml and centrifuged at 1,200 *g* for 8 min. The cell suspension was mixed 1:10 with 0.75% low melting point agarose at 37°C, dropped on GelBond® Films (GBF) (Life Sciences, Lithuania) in triplicates and lysed in cold lysis buffer overnight at 4°C. GBF were then incubated in electrophoresis buffer, to allow DNA denaturation and expression of alkali-labile sites, for 35 min at 4°C. Subsequently, the electrophoresis step was carried out at 20 V and 300 mA for 20 min at 4°C. GBF were washed twice with cold PBS 1× fixed in absolute ethanol for 1 h, then air-dried overnight at room temperature. Cells were stained with 1:10,000 SYBR Gold in TE buffer for 20 min at room temperature, mounted, and visualized with an epifluorescence microscope (Olympus BX50) with a 20× magnifications. DNA damage was quantified with the Komet 5.5 Image analysis system (Kinetic Imaging Ltd, Liverpool, UK) as the percentage of DNA in the tail. One hundred randomly selected comet images were analyzed per sample (33/34 per triplicate). Results from three independent biological replicates were analyzed.

### Targeted liquid chromatography-mass spectrometry (LC-MS) analyses

Metabolites were extracted from snap-frozen cell pellets by adding methanol: $H_2O$ (1:1 v/v) solution and vortexing samples for 30 s. Samples were immersed in liquid $N_2$ to disrupt cell membranes followed by 30 s of ultrasonication. These two steps were repeated

three times. Then, samples were incubated at −20°C for 2 h, centrifuged at 17,000 × *g* for 15 min, and the supernatant was collected into a LC-MS vial. Samples were injected in a UHPLC system (1290 Agilent) coupled to a triple quadrupole (QqQ) MS (6490 Agilent Technologies) operated in multiple reaction monitoring (MRM) and positive (POS) or negative (NEG) electrospray ionization (ESI) mode. Metabolites were separated using C18-RP (ACQUITY UPLC BEH C18 1.7 μM, Waters) chromatography at flow rate of 0.3 ml/min. The solvent system was A = 20 mM ammonium acetate + 15 mM $NH_3$ in water and B = acetonitrile: water (95:5). The linear gradient elution started at 100% A (time 0–2 min), 65% A (time 2–5 min) and finished at 100% B (time 5.5 min). MRM transitions are shown in Appendix Table S3.

### Statistical analysis

Statistical analysis was performed using GraphPad Prism version 7.00 (GraphPad Software, San Diego, CA, USA). Data are presented as mean ± SD. Experimental datasets were compared by (i) two-sampled, two-tailed Student's *t*-test to compare two experimental conditions sharing normal distribution and variance and (ii) non-parametric Kruskal–Wallis test for more than two conditions. Multiple comparisons were corrected using Tukey's test for equal variances or using Dunett's T3 test for different variances.

## Data availability

This study includes no data deposited in external repositories.

**Expanded View** for this article is available online.

## Acknowledgements

We thank past and present members of the Laboratory of Cancer Metabolism at IDIBELL-ICO, the Department of Internal Medicine at the University of Cincinnati for sharing ideas and reagents as well as their encouragement throughout the study, and the CERCA Programme/Generalitat de Catalunya for institutional support. We thank Dr. J. Bartek for his critical comments of earlier drafts of this manuscript. We also thank Drs. S. Fumagalli and S. Volarevic for their advice during these studies, Drs. B. Vogelstein and K. Vousden that kindly provided isogenic HCT116 p21$^{-/-}$ cells, and Dr B.E. Sawaya that kindly provided the pcDNA3.1-p21 plasmid. These studies were supported by grants to G. T. from the Spanish Ministry of Economy and Competitiveness (SAF2014-52162-P), the CIG European Commission (PCIG10-GA-2011-304160), the NIH/NCI National Cancer Institute (R01-CA191814), the ISCIII-RTICC (RD12/0036/0049), the AGAUR (2014SGR-870) and by shared funding from the IDIBELL and the Vall d'Hebron Institute of Oncology (VHIO). A. G. and G. T. also were supported by a grant from the Spanish Ministry of Economy and Competitiveness (SAF2017-84301-P) and by the European Social Fund (ESF) "Investing in your future" (AGAUR—2017 SGR 01743). R. S. and G.T. were supported by a grant from the Asociación Española Contra el Cáncer (GCB-142035THOM). N.A. was supported by a grant from the Spanish Ministry of Economy and Competitiveness SAF2016-76239-R. The Spanish Ministry grants to G. T. and A. G. are co-funded by FEDER—a way to build Europe. J. P. was supported by fellowships from the Association pour la recherche sur le Cancer (SAE20140601346) and the Juan de la Cierva (FJCI-2014-20422). C.M. was supported by European Union's Horizon 2020 research and innovation program under the Marie Sklodowska-Curie grant agreement (M-Lysosomes, 799000).

## Author contributions

JP, FR-C and GT conceived and designed the study. JP and FR-G performed most of the experiments, and EA, CM, SS, SF, SM, AD, MG-C, CC and AG contributed to individual experiments. JP, OY, NA, RM, RS, AT, SCK, AG, and GT analyzed the data, and all authors provided intellectual support in the discussion of the results. JP and GT wrote the manuscript.

## Conflict of interest

The authors declare that they have no conflict of interest.

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
