## [Review Process File · The EMBO Journal]

Nucleotide depletion reveals the impaired ribosome biogenesis checkpoint as a barrier against DNA damage

Joffrey Pelletier, Ferran Riaño-Canalias, Eugènia Almacellas, Caroline Mauvezin, Sara Samino, Sonia Feu, Sandra Menoyo, Ana Domostegui, Marta Garcia-Cajide, Ramon Salazar, Constanza Cortés, Ricard Marcos, Albert Tauler, Oscar Yanes, Neus Agell, Sara C.Kozma, Antonio Gentilella and George Thomas

Review timeline:

Submission date:	26 October 2019
Editorial Decision:	27 November 2019
Revision received:	7 March 2020
Additional Correspondence:	3 April 2020
Additional Correspondence:	11 April 2020
Accepted:	14 April 2020

Editor: Hartmut Vodermaier

Transaction Report:

1st Editorial Decision

27 November 2019

Thank you for submitting your full manuscript to our editorial office. We had sent it to three expert reviewers, who have now returned the reports copied below. As you will see, the referees acknowledge the potential interest of your study, but especially referees 2 and 3 also raise a number of substantive issues that would need to be decisively addressed before publication may be warranted.

Although these referee concerns have the potential to undermine key conclusions of the study, I would -given that the concerns are quite specific- nevertheless like to give you an opportunity to address them by way of a revised manuscript. For such a revision to be successful, the main points would be

- 1) Decisively putting the presented 'hierarchical' model to the tests as requested by referee 2, and if necessary revising it (see referee 2)
- 2) More convincingly assessing the role/presence of replication stress and associated parameters (e.g. fork degradation, RPA foci) (see referee 3)
- 3) Improving quantitations and statistical analysis of all data, and better describing experimental procedures and rationales

Should you be able to satisfactorily address the above-listed main issues as well as the various more minor/specific points raised by all three reviewers, we would be happy to consider a revised manuscript further for EMBO Journal publication. I realize that this may require considerable further time and effort, and would therefore be open to discussing a possible extension of the default three-months revision deadline, during which publication of any competing/related work would as usual not have a negative impact on our final assessment of your own study. Please be reminded that our policy to allow only a single round of major revision will make it important to comprehensively answer to all points raised at this stage. I would further encourage you to contact me already during

the early stages of revision to discuss any proposals for addressing the reviewers' concerns

Further information on preparing and uploading revised manuscript files can be found below and in our Guide to Authors.

Thank you again for the opportunity to consider this work for The EMBO Journal. I look forward to hearing from you in due time.

REFeree REPORTS

Referee #1:

The manuscript by Dr. Thomas and colleagues, entitled "Nucleotide depletion reveals the Impaired Ribosome Biogenesis Checkpoint as a barrier against DNA damage", clearly demonstrates that nucleotide depletion controls p53 activation, in a hierarchical manner, by the Impaired Ribosome Biogenesis Checkpoint (IRBC) and the DNA Damage response (DDR). They find that the inhibition of inosine monophosphate dehydrogenase (IMPDH) activates the IRBC as a primary sensor, causing p21-mediated G1-arrest. They also show that prolonged IMPDH inhibition, however, leads to p21 degradation, allowing S-phase entry and DDR activation, delaying S phase progression. Their data is clean and straightforward, the text is clear, and the conclusions are well supported by the experiments. Therefore I consider that only a few experiments are required before acceptance for publication.

Major points:

Fig. 1: I recommend the authors to show the effect of MPA on the nucleolar structure because IRBC induced- p53 activation is tightly regulated by 5S RNP whose localization is tightly regulated by structural change of the nucleolus.

Fig. 1A and Fig. S1C: They should show the positive control for P-Chk2 (T68) and γ -H2AX, marker of DNA double strand break (for example, etoposide +).

Fig. 1D: They should show that IMPDH1 protein is reduced by siRNA for IMPDH1 using IMPDH1 antibody. It is possible that residual IMPDH1 may be cause of lack of P-Chk1 (S345).

Minor points:

Fig. S1B. Is there any difference in 53BP1 staining between control cells and MPA treated cells? For me, it looked similar.

Fig. S2C: They described that the ATR inhibitor induced phosphorylation of P-Chk2 (T68) and γ -H2AX, marker of ATM activation, an effect of suppressed by ATM inhibitor (page 9; lines 27-30). However, it looks for me that the intensity of the γ -H2AX band is not reduced by the ATM inhibitor. The author should comment about that.

Fig. 5A (right top graph) and Fig. S6A: The summation of percentage of the G0/G1, S, and G2/M phase cells is not 100% (Fig 5A, bar 3; Fig. S6A, bar 4). They must explain the reason.

Page 11, last line: "but all having a functional p53 and (Fig. 4B and s4D)." p53 and what?

Referee #2:

This interesting study provides solid in vitro data for impaired ribosome biogenesis checkpoint (IRBC) as the primary response to nucleotide depletion with the IMPDH inhibitor MPA. The study also examines a distinct secondary input to p53 induced by replication stress when inhibition of GMP synthesis is more severe. While the data is intriguing, the hierarchical organization of the two checkpoints requires validation.

The model proposed implicates p21 expression dynamics in mediating response to nucleotide depletion. A reduction in dGTP pool led to activation of IRBC and induced G1 arrest. Severe nucleotide depletion however was shown to be associated with a compromised G1 checkpoint due to reduced p21 levels. The authors provide data demonstrating that p21 is required for the G1/S checkpoint following 10 μ M MPA.

This model should incorporate recent findings in a high profile paper in Cell (Chien-Hsiang Hsu et al., 2019 PMID 31204100) that demonstrated p21 levels to be induced by ATM in G1 and repressed by CHK1 in S phase in response to non-lethal doses of chemotherapy. Hsu et al identified delayed-low p21 levels in S/G2 of drug treated cells to be associated with senescence fate. Using genetic and pharmacological perturbations, Chk1 activity and proteasomal degradation were identified as molecular mechanisms that repressed p21 expression in S phase. Indeed, in Pelletier et al, induced CHK1 activity in response to IMPDH inhibition seems to correlate with repressed p21 expression.

Therefore, it is plausible that under conditions of high IMPDH inhibition, global replication stress can be the dominant stress pathway leading to induced CHK1 activity mediating repression of p21 levels. ie while IRBC serves as a sensor for reduced nucleotide pool, the DDR is the major checkpoint that compromises IRBC via repressing p21 levels to mediate S/ G2 arrest. Therefore, the model may not be hierarchal as described in the manuscript. Moreover, this non-hierarchal model may be in agreement with the Valvezan et al study which the authors discuss.

Major points:

The data in figure 2 demonstrate that IRBC and ATR distinctly regulate p53 stabilization by MPA. However, the data has some limitations.

- The data in Figure 2A show that inhibiting the IRBC by depleting RPL11 does not completely prevent induction of p53 levels by high concentration (10 μ M) of MPA. Why was this experiment performed at 6hours post treatment? This experiment should also be performed with 1 μ M MPA where maximum induction of p21 is expected. It would co-depleting RPL11 and RPL7 completely prevent p21 induction following low IMPDH inhibition? This experiment will validate IRBC as the major as the primary sensor for reduced dGTP pool by low dose MPA.

- As mentioned above, CHK1 can inhibit p21 expression (Chien-Hsiang Hsu et al., 2019). CHK1 signalling and proteasomal degradation led to reduced p21 expression in S/G2 cells treated with low dose chemotherapy. Indeed, data in Fig S2C demonstrates co-treatment of ATRi and MPA to be associated with an increase in p21 levels. Furthermore, Chien-Hsiang Hsu et al demonstrated ATM signalling to be required for inducing p21 expression in G1 cells. Therefore, Figure 2C should be repeated with inhibiting ATM and ATR or CHK1 individually in RPL11 depleted cells treated with MPA. Ideally, assessment of p21, pCHK1 and pCHK2 levels should be performed in cells isolated at different stages of the cell cycle or in synchronised cell populations.

Minor points

1. Quantitation of immunofluorescence data in Fig S1B should be included.
2. Page 17 "... the development of new drugs specifically designed to attack ribosome biogenesis, currently in clinical trials for specific cancers (Bywater et al, 2012; Drygin et al, 2011)." The Phase I clinical trial of CX-5461 was recently published in Cancer Discovery and should be referenced.

Referee #3:

In this manuscript, Pelletier and colleagues address the relationship between the impaired ribosome biogenesis checkpoint (IRBC) and the DNA damage response (DDR) upon depletion of nucleotides. They first show that gradual inhibition of inosine monophosphate dehydrogenase (IMPDH), an enzyme required for GMP synthesis, leads to cell cycle arrest due to p53 activation and p21 accumulation. Surprisingly, at higher doses of the IMPDH inhibitor MPA, p21 levels drop and cells re-enter S-phase. In addition, high levels of MPA also induce gammaH2AX phosphorylation at later time points. The authors then show that inactivation of the IRBC has a similar effect even in the absence of nucleotide depletion, which they claim is the result of replicative stress followed by replication fork collapse; but in my opinion, this is an overinterpretation of the data (see below). In summary, I do not think that this manuscript offers enough evidence to support the main

conclusions. In addition, the data are partially not presented in an informative manner and statistical analysis is also partially done incorrectly (see specific points for details). I therefore do not feel that this paper is suitable for publication in EMBO J. in its current form.

Specific points:

1) Many results include quantification of Western blots by densitometry. The results are then represented by bar graphs with error bars and statistical significance is tested with two-tailed t-tests. The problem is that information on the exact sample size is missing in most cases. As bar graphs are often misleading, a better way to visualize these data would be to show the individual data points. That would give a good representation of the spread of the data and would be more helpful for the interpretation than bar graphs with error bars. Also, I don't think that with a sample size of only 2 (Figure 2C), error bars and t-tests make any sense.

2) The rationale of the fiber assay in Figure 4D and E is not clear to me. It seems that the drug was applied 2h or 23h before the fiber labeling. But then only IdU track length were measured. Why is that? That does not make any sense to me in this context. Why were two different nucleotide analogs used in the first place? Also, I don't see why two analogous representations of the results were given in Figure 4E. Both show distribution of the IdU track length. It would have been more informative if fork asymmetry was assessed, which would have been possible with the two nucleotide analogs labeling protocol. Also, since the authors claim that MPA treatment leads to fork instability and fork collapse, the shorter IdU track length measured upon treatment with 10 μ M MPA could also reflect fork degradation and not fork stalling.

3) The assays performed in Figure 6 are not supporting the conclusions drawn by the authors. They claim that inactivation of the IRBC allows cells to enter S-phase with low nucleotide levels where they encounter replication stress and DNA damage. However, the assays performed only assess DNA damage (gammaH2AX and 53BP1) but not replication stress. For example, if the DSBs are the result of collapsed replication forks, they should only occur in S-phase cells. However, the data in Figure 6C-E do not indicate if the DNA damage occurs in S-phase cells.

4) GammaH2AX and 53BP1 staining on their own are not good enough to indicate the presence of DNA double-strand breaks (DSBs), especially when they don't show a focal staining pattern. Either pulsed-field gel electrophoresis or neutral comet assay should be done to support the claim that MPA treatment or inactivation of the IRBC leads to DSB break formation in S-phase. To directly show that MPA treatment leads to DSB formation at collapsed replication forks, EdU staining on metaphase chromosomes should be performed to show that the EdU signal is localized predominantly at sites with hallmarks of chromosome breaks such as terminal parts of shattered chromosomes or boundaries between fused chromosome fragments (see e.g. Toledo et al. 2013, Cell 155, 1088-1103)

5) What is generally lacking is solid evidence for DNA replication stress in the presence of high concentrations of MPA and/or inactivation of the IRBC. The fiber assay in Figure 4 might suggest replication stress but as mentioned above, could also indicate increased fork degradation. EdU incorporation measurements could be a way to test for replication stress. Nucleotide depletion was also shown to lead to ssDNA accumulation. RPA foci formation or (even better) BrdU staining under non-denaturing conditions could indicate the presence of increased ssDNA. This should be done to support the claim that there is increased replication stress in these cells.

6) In Figure 6C and 6E, two tailed t-tests were used to test for statistical significance. This is not the correct statistical analysis as t-tests are parametric tests that only deliver reliable results on normally distributed data. The graphs in Figure 6C and D clearly show that the data is not normally distributed. Moreover, how were the multiple t-test corrected for alpha-error propagation? When more than two groups are compared, ANOVA is a better statistical test, but it also assumes normally distributed data.

Referee#1:

The manuscript by Dr. Thomas and colleagues, entitled "Nucleotide depletion reveals the Impaired Ribosome Biogenesis Checkpoint as a barrier against DNA damage", clearly demonstrates that nucleotide depletion controls p53 activation, in a hierarchical manner, by the Impaired Ribosome Biogenesis Checkpoint (IRBC) and the DNA Damage response (DDR). They find that the inhibition of inosine monophosphate dehydrogenase (IMPDH) activates the IRBC as a primary sensor, causing p21-mediated G1-arrest. They also show that prolonged IMPDH inhibition, however, leads to p21 degradation, allowing S-phase entry and DDR activation, delaying S phase progression. Their data is clean and straightforward, the text is clear, and the conclusions are well supported by the experiments. Therefore I consider that only a few experiments are required before acceptance for publication.

We were pleased that the referee found the present manuscript "clean and straightforward", though he/she requested a number of experiments before acceptance.

Major points:

#1 - Fig. 1: I recommend the authors to show the effect of MPA on the nucleolar structure because IRBC induced- p53 activation is tightly regulated by 5S RNP whose localization is tightly regulated by structural change of the nucleolus.

We thank the referee for this suggestion. In the revised version of the manuscript we show that increasing doses of MPA lead to the disruption of the nucleolus. Similar findings are also shown for ActD, similar to those we previously published (Fumagalli *et al.*, 2012. *Genes & development* 26, 1028-1040). Disruption of the nucleolus by either MPA or ActD is evidenced by the redistribution of upstream binding factor (UBF) and fibrillarin to nucleolar cap structures (see pg. 10 and new Fig S3C)

#2 - Fig. 1A and Fig. S1C: They should show the positive control for P-Chk2 (T68) and γ -H2AX, marker of DNA double strand break (for example, etoposide +).

We had included etoposide in the original blot for Figure 1A, but removed it later, assuming that ActD was a sufficient control. In the revised version of the manuscript we have included a comparison of the effects of MPA and AVN versus that of etoposide on Chk2 T68 phosphorylation and γ -H2AX, demonstrating in the latter case a strong induction of both responses (see pg. 7 and new Fig S1F).

#3 - Fig. 1D: They should show that IMPDH1 protein is reduced by siRNA for IMPDH1 using IMPDH1 antibody. It is possible that residual IMPDH1 may be because of lack of P-Chk1 (S345).

These are valid points, however, IMPDH1 and IMPDH2 share 84% identity and despite the claims of the producers, we found no commercial antibodies, which specifically detected IMPDH1 without cross reacting with IMPDH2. The so-called "IMPDH1 antibodies" were generated using the full length IMPDH1 protein or smaller epitopes, in the latter case they still presented a high degree of homology with IMPDH2. However, when we used shRNAs to deplete IMPDH1, we saw no changes on Western blots, despite depleting its cognate mRNA to less than 20% of its original levels. In contrast, if we depleted IMPDH2, we lost the detection of both IMPDH1 and 2, by the so-called "IMPDH1 antibodies" and so we have referred to this antibody as a PAN IMPDH antibody. We have explained this point to some extent in the Material and Methods section of the revised manuscript (see pg. 21). As regards having "residual IMPDH1" present, we wrote in the original manuscript that having residual IMPDHs following siRNA depletion was most likely responsible for the lack of replicative stress and Chk1 S345 phosphorylation (pg. 8), consistent with higher levels of guanosine nucleotides in IMPDH depleted cells versus those treated with MPA (Fig S1M and N).

Minor points:

#1 - Fig. S1B. Is there any difference in 53BP1 staining between control cells and MPA treated cells? For me, it looked similar.

We have quantitated this data and added new representative confocal images in the revised manuscript (see new Fig S1B-D), as also suggested by Referee 2, Minor point 1. The results show no significant differences in the number of γ -H2AX or 53BP1 foci in cells treated for 24 hrs with MPA (pg. 7).

#2 - Fig. S2C: They described that the ATR inhibitor induced the phosphorylation of P-Chk2 (T68) and γ -H2AX, marker of ATM activation, an effect of suppressed by ATM inhibitor (page 9; lines 27-30). However, it looks for me that the intensity of the γ -H2AX band is not reduced by the ATM inhibitor. The author should comment about that.

We thank the referee for bringing up this point. Other kinases are activated by DNA damage, which can phosphorylate H2AX, most likely DNA-PK. We have rewritten this section, in the revised manuscript, to include this point (see pgs. 9 and 10).

#3 - Fig. 5A (right top graph) and Fig. S6A: The summation of percentage of the G0/G1, S, and G2/M phase cells is not 100% (Fig 5A, bar 3; Fig. S6A, bar 4). They must explain the reason.

We apologize for this oversight, we simply rounded off the FACS data to the nearest whole number, which did not come out to 100%. We have corrected this oversight in the revised manuscript.

Page 11, last line: "but all having a functional p53 and (Fig. 4B and s4D)." p53 and what?

This was a last minute typo, the "and" has been deleted.

Referee #2:

This interesting study provides solid in vitro data for impaired ribosome biogenesis checkpoint (IRBC) as the primary response to nucleotide depletion with the IMPDH inhibitor MPA. The study also examines a distinct secondary input to p53 induced by replication stress when inhibition of GMP synthesis is more

severe. While the data is intriguing, the hierarchical organization of the two checkpoints requires validation.

We are pleased that the reviewer found this an interesting paper. We have addressed his/her points as described below.

The model proposed implicates p21 expression dynamics in mediating response to nucleotide depletion. A reduction in dGTP pool led to activation of IRBC and induced G1 arrest. Severe nucleotide depletion however was shown to be associated with a compromised G1 checkpoint due to reduced p21 levels. The authors provide data demonstrating that p21 is required for the G1/S checkpoint following 10 μ M MPA.

We thank the referee for his/her comment but we would like to clarify a potential misunderstanding as we show that the depletion of guanine ribonucleotides leads to the activation of the IRBC at 1 μ M MPA, but that dGTP levels only fall at higher concentrations, i.e. 10 μ M MPA (Fig 1B), when we detect Chk1 activation and a reduction in the p21 protein response (Fig 1A).

This model should incorporate recent findings in a high profile paper in Cell (Chien-Hsiang Hsu et al., 2019 PMID 31204100) that demonstrated p21 levels to be induced by ATM in G1 and repressed by CHK1 in S phase in response to non-lethal doses of chemotherapy. Hsu et al identified delayed-low p21 levels in S/G2 of drug treated cells to be associated with senescence fate. Using genetic and pharmacological perturbations, Chk1 activity and proteasomal degradation were identified as molecular mechanisms that repressed p21 expression in S phase. Indeed, in Pelletier et al, induced CHK1 activity in response to IMPDH inhibition seems to correlate with repressed p21 expression.

Please see response to Major point 2 below.

Therefore, it is plausible that under conditions of high IMPDH inhibition, global replication stress can be the dominant stress pathway leading to induced CHK1 activity mediating repression of p21 levels. ie while IRBC serves as a sensor for reduced nucleotide pool, the DDR is the major checkpoint that compromises IRBC via repressing p21 levels to mediate S/ G2 arrest. Therefore, the model may not be hierarchal as described in the manuscript. Moreover, this non-hierarchal model may be in agreement with the Valvezan et al study which the authors discuss.

As regards the hierarchal response, based on the paradox between the Lu and Dang laboratories and our findings in Figure 1A, we chose to compare 1 and 10 μ M MPA for their ability to induce the IRBC and the DDR. This decision was built on the hypothesis that the IRBC prevents S-phase entry, and as a consequence would inhibit replicative stress and the activation of the ATR-Chk1 pathway. The hierarchy of the IRBC and ATR-Chk1 checkpoints stems from the differential sensitivity of activation of these responses and from the dependency of the two checkpoints on one another. The latter point is based on the depletion of RPL11 (Fig 6A), which like loss of p21 (Fig S5A) or p53 (Fig 6A) increases S-phase entry induced by 10 μ M MPA, ultimately leading to increased risk of replicative stress. It should also be pointed out that the results of Valvezan *et al.*, that inhibition of IMPDH has no effect on ribosome biogenesis, disagrees with many papers in the literature, as we point out in the Discussion (pg. 18).

Major points:

#1 - The data in figure 2 demonstrate that IRBC and ATR distinctly regulate p53 stabilization by MPA. However, the data has some limitations.

• The data in Figure 2A show that inhibiting the IRBC by depleting RPL11 does not completely prevent induction of p53 levels by high concentration (10 μ M) of MPA. Why was this experiment performed at

6 hours post treatment? This experiment should also be performed with 1 μ M MPA where maximum induction of p21 is expected. It would co-depleting RPL11 and RPL7 completely prevent p21 induction following low IMPDH inhibition? This experiment will validate IRBC as the major as the primary sensor for reduced dGTP pool by low dose MPA.

The experiment performed in Figure 2A was carried out after a 24 hr, not a 6 hr treatment, with either 1 or 10 μ M MPA. The experiment shows that if we first deplete cells of RPL11, we observe higher levels of p53 in the presence of 10 μ M MPA. This along with the data in Figure 1A, suggested the existence of a second checkpoint. In Figure 2B, we conducted the co-depletion of RPL7a and RPL11. The aim of this experiment was to confirm that the conditions used for the depletion of RPL11 were optimal in preventing p53 induction by the IRBC, as the depletion of RPL7a specifically activates the IRBC, leading to the induction of p53, which can be completely reversed by co-depleting RPL11. In contrast, if we add MPA after co-depletion of RPL7a and RPL11, p53 is still induced.

#2 - • As mentioned above, CHK1 can inhibit p21 expression (Chien-Hsiang Hsu et al., 2019). CHK1 signalling and proteasomal degradation led to reduced p21 expression in S/G2 cells treated with low dose chemotherapy. Indeed, data in Fig S2C demonstrates co-treatment of ATRi and MPA to be associated with an increase in p21 levels. Furthermore, Chien-Hsiang Hsu et al demonstrated ATM signalling to be required for inducing p21 expression in G1 cells. Therefore, Figure 2C should be repeated with inhibiting ATM and ATR or CHK1 individually in RPL11 depleted cells treated with MPA. Ideally, assessment of p21, pCHK1 and pCHK2 levels should be performed in cells isolated at different stages of the cell cycle or in synchronised cell populations.

We thank the referee for pointing out the Chien-Hsiang Hsu et al work, as we were unaware of its publication. In Figure 2C, we show that selective ATR inhibition, alone or in combination with MPA, blocks Chk1 phosphorylation, which is associated with the induction of p21. However as pointed out in our paper, ATR protects the stability of replicative forks, such that its inhibition leads to ATM-mediated p53^{S15}, Chk2^{T68} and H2AX phosphorylation (pgs. 9 and 10). Thus, the effects on p21 could be due to DNA damage and activation of the DDR rather than inhibition by Chk1, with cells eventually entering into senescence. We should also add, that the synchronization experiment shown in Figure 5B, demonstrates that the loss of p21 is necessary for S-phase entry. This result suggests that Chk1 activation, which requires replicative stress, may not be causative in the degradation of p21 and the alterations of the G1/S checkpoint. In the Discussion of the revised manuscript we comment on Chien-Hsiang Hsu et al. findings, and on potential mechanisms involved in the loss of p21 (pgs. 16 and 17)

Minor points

1. Quantitation of immunofluorescence data in Fig S1B should be included.

We agree with the referee. Please see response to Referee 1, Minor point 1.

2. Page 17 "... the development of new drugs specifically designed to attack ribosome biogenesis, currently in clinical trials for specific cancers (Bywater et al, 2012; Drygin et al, 2011)." The Phase I clinical trial of CX-5461 was recently published in Cancer Discovery and should be referenced.

The referee is correct and we have included this reference in the revised manuscript (see pg. 18)

Referee #3:

In this manuscript, Pelletier and colleagues address the relationship between the impaired ribosome biogenesis checkpoint (IRBC) and the DNA damage response (DDR) upon depletion of nucleotides. They

first show that gradual inhibition of inosine monophosphate dehydrogenase (IMPDH), an enzyme required for GMP synthesis, leads to cell cycle arrest due to p53 activation and p21 accumulation. Surprisingly, at higher doses of the IMPDH inhibitor MPA, p21 levels drop and cells re-enter S-phase. In addition, high levels of MPA also induce gammaH2AX phosphorylation at later time points. The authors then show that inactivation of the IRBC has a similar effect even in the absence of nucleotide depletion, which they claim is the result of replicative stress followed by replication fork collapse; but in my opinion, this is an overinterpretation of the data (see below).

We thank the referee for bringing out this point, as our intention here was to emphasize that induction of DNA damage in cells depleted for RPL5/RPL11/TIF3A, in absence of nucleotide depletion, was unexpected (pg. 14). We agree with the reviewer that the mechanism involved under these conditions is most likely distinct from the enhanced replicative stress induced by nucleotide depletion. Indeed in the Discussion of the original manuscript we proposed two alternative mechanisms, i.e the downregulation of genes involved in the DDR, or the decreased translation of factors necessary for correct DNA replication (pg. 17 and 18). In the revised manuscript we have further clarified this point (see pg. 15).

In summary, I do not think that this manuscript offers enough evidence to support the main conclusions. In addition, the data are partially not presented in an informative manner and statistical analysis is also partially done incorrectly (see specific points for details). I therefore do not feel that this paper is suitable for publication in EMBO J. in its current form.

We hope that the revisions and additional experiments proposed below will convince the referee of the importance and timeliness of our findings.

Specific points:

1) Many results include quantification of Western blots by densitometry. The results are then represented by bar graphs with error bars and statistical significance is tested with two-tailed t-tests. The problem is that information on the exact sample size is missing in most cases. As bar graphs are often misleading, a better way to visualize these data would be to show the individual data points. That would give a good representation of the spread of the data and would be more helpful for the interpretation than bar graphs with error bars. Also, I don't think that with a sample size of only 2 (Figure 2C), error bars and t-tests make any sense.

Almost all the experiments have been performed at least three independent times. As suggested by the referee, in the revised manuscript we have systematically shown individual data points for western-blot quantifications and re-analyzed the statistical significance of the results.

2) The rationale of the fiber assay in Figure 4D and E is not clear to me. It seems that the drug was applied 2h or 23h before the fiber labeling. But then only IdU track length were measured. Why is that? That does not make any sense to me in this context. Why were two different nucleotide analogs used in the first place?

We thank the referee for this comment, as we obviously did not explain this experiment sufficiently. Even though measuring replicative fork speed can be performed by following the incorporation of a single thymidine analog, we consider that the findings are more robust if two analogs are employed sequentially, which is also a standard methodology described by others (Bester et al., 2011. Cell 145, 435-446). We initially pulse with the first analog (CldU here), to mark the fibers that are elongating and add the second analog (IdU here) to measure the track length. We only measured those fibers that are doubly labeled, to

ensure that we are not measuring fibers that were initiated during the second labeling period, which would give a shorter length. This has been clarified in the Materials and Methods section (pgs. 24 and 25).

.Also, I don't see why two analogous representations of the results were given in Figure 4E. Both show distribution of the IdU track length.

We agree with the referee. One representation has been removed in the revised manuscript.

It would have been more informative if fork asymmetry was assessed, which would have been possible with the two nucleotide analogs labeling protocol.

We thank the referee for his/her excellent suggestion. We show in the revised version of the manuscript that the reduced rate of DNA fiber elongation is associated with asymmetric replication of the two DNA strands (see new Fig 4C, right panel).

Also, since the authors claim that MPA treatment leads to fork instability and fork collapse, the shorter IdU track length measured upon treatment with 10 μ M MPA could also reflect fork degradation and not fork stalling.

Here we would like to point out that we are able to detect incorporation of the two analogs in a 30 min time frame, and that we show by propidium iodide DNA staining that the DNA slowly replicates in a 72 hr time course after G1-synchronization (Fig S4C). Therefore, even though, we do not discard the possibility that fork degradation may occur in MPA treated cells, the net balance between DNA synthesis and degradation favors synthesis. Moreover, we measured DNA fork speed to support the concept that MPA treatment leads to replicative stress. As replication fork degradation often occurs as a consequence of replicative stress (Quinet, et al., 2017. Molecular Cell 68, 830-833), the possibility that fork degradation may be increased with MPA does not affect our conclusions.

3) The assays performed in Figure 6 are not supporting the conclusions drawn by the authors. They claim that inactivation of the IRBC allows cells to enter S-phase with low nucleotide levels where they encounter replication stress and DNA damage. However, the assays performed only assess DNA damage (gammaH2AX and 53BP1) but not replication stress. For example, if the DSBs are the result of collapsed replication forks, they should only occur in S-phase cells. However, the data in Figure 6C-E do not indicate if the DNA damage occurs in S-phase cells.

The referee is correct. To address whether DNA damage occurs as a consequence of replicative stress in MPA-treated cells, we followed cell cycle progression and γ H2AX levels by cytometry, after G1 synchronization and release for increasing time with MPA. Critically, we show in the revised manuscript that MPA-treated cells acquire DSBs when they are exclusively in S-phase (Fig S4J). We also briefly clarify in the introduction that MPA, unlike nucleoside analogs that produce chromosome breaks or that inhibit DNA repair enzymes, is a catalytic non-competitive inhibitor of IMPDH, which does not incorporate into the DNA (pg 5), probably acting through the inhibition of dGTP synthesis. Therefore, considering that (1) cells replicating DNA with MPA encounter DNA damage (Fig S4J) and (2) depletion of IRBC complex components increases the proportion of cells in S-phase with MPA (Fig 6A), it is likely that these cells acquire DNA damage in response to increased replicative stress. We have clarified this point in the result section (pg. 15).

4) GammaH2AX and 53BP1 staining on their own are not good enough to indicate the presence of DNA double-strand breaks (DSBs), especially when they don't show a focal staining pattern. Either pulsed-

field gel electrophoresis or neutral comet assay should be done to support the claim that MPA treatment or inactivation of the IRBC leads to DSB break formation in S-phase. To directly show that MPA treatment leads to DSB formation at collapsed replication forks, EdU staining on metaphase chromosomes should be performed to show that the EdU signal is localized predominantly at sites with hallmarks of chromosome breaks such as terminal parts of shattered chromosomes or boundaries between fused chromosome fragments (see e.g. Toledo et al. 2013, Cell 155, 1088-1103)

The referee is correct. In the revised manuscript we performed the alkaline comet assay to detect fragmented DNA, which is considered more sensitive for smaller amount of DNA damage than the neutral comet assay (Olive et al., 2006. Nature Protocols 1, 23-29). We detected a two-fold increase in the percentage of tail DNA / total DNA, for both loss of the IRBC or treatment with MPA, an affect which is augmented when cells are depleted of the IRBC and then treated with MPA (see new Fig 6F). Even though performing EdU staining on metaphase chromosomes would be of interest, this appears to go beyond the scope of this study.

5) What is generally lacking is solid evidence for DNA replication stress in the presence of high concentrations of MPA and/or inactivation of the IRBC. The fiber assay in Figure 4 might suggest replication stress but as mentioned above, could also indicate increased fork degradation. EdU incorporation measurements could be a way to test for replication stress. Nucleotide depletion was also shown to lead to ssDNA accumulation. RPA foci formation or (even better) BrdU staining under non-denaturing conditions could indicate the presence of increased ssDNA. This should be done to support the claim that there is increased replication stress in these cells.

As proposed by the referee we show in the revised manuscript that MPA increases the generation of ssDNA detected by BrdU staining under non-denaturing conditions and accumulation of chromatin-loaded RPA (Fig 4E and S4H). We consider that DNA fork elongation inhibition and asymmetric progression, the presence of ssDNA, and activation of ATR-CHK1, are sufficient to support that MPA leads to replicative stress.

6) In Figure 6C and 6E, two tailed t-tests were used to test for statistical significance. This is not the correct statistical analysis as t-tests are parametric tests that only deliver reliable results on normally distributed data. The graphs in Figure 6C and D clearly show that the data is not normally distributed. Moreover, how were the multiple t-test corrected for alpha-error propagation? When more than two groups are compared, ANOVA is a better statistical test, but it also assumes normally distributed data.

We agree with the referee and apologize for this oversight. Test of normality has been performed and the appropriate statistical test has been used accordingly.

Additional Correspondence

3 April 2020

Thank you for submitting your revised manuscript for our consideration. It has now been seen again by the three original referees, and I am pleased to say that all of them are largely satisfied with your revision and have no more reservations towards publication in The EMBO Journal. As you will see from the comments below, there are only a few minor textual and presentational issues remaining, which I would like to ask you to incorporate into the text (and figures if necessary) during a final minor revision.

REFeree REPORTS

Referee #1 (Report for Author)

The author improved manuscript in response to my comment and their responses were reasonable. Therefore, I consider that this manuscript should be accepted.

Referee #2 (Report for Author)

The manuscript is improved. The authors have addressed my concerns.

Minor points:

The new quantitation of the fibre assay data in 4C Fig can be confusing on its own as it can also indicate fork progression (with the second track being longer than the 1st track under treatment conditions). The authors should include quantitation of the length of the second track only to show that the track length is shorter following treatment compared to control, which is more representative of the images shown in Fig 4E

Referee #3 (Report for Author)

The authors have addressed most of my previous questions and concerns. When possible, they also performed additional experiments to support their conclusions. Some of them (for example the replication fork asymmetry assay) allowed very interesting new insights. Unfortunately, a few important issues were not yet addressed adequately in my opinion. For example, alkaline comet assay is not appropriate to indicate the presence of DNA double-strand breaks, because when the strands are separated by the alkaline conditions, mostly single-strand nicks and gaps are revealed by this assay. Neutral comet assays should have been performed, as I suggested in my first review. Overall though, I think the authors did a decent job answering to the reviewers comments. Given the current international Corona crisis, where most labs (including my own) had to temporarily close down, I think it is a responsible position not to ask further experimental evidence from the authors, but only textual revisions. One such textual revision that I strongly suggest is the clear indication that the alkaline comet assay data in Fig 6F reveal predominantly single-strand nicks and gaps, but not double-strand breaks.

Additional Correspondence

11 April 2020

Referee #1 (Report for Author)

The author improved manuscript in response to my comment and their responses were reasonable. Therefore, I consider that this manuscript should be accepted.

We are pleased the referee considered we have addressed his/her concerns.

Referee #2 (Report for Author)

The manuscript is improved. The authors have addressed my concerns.

We are pleased the referee considered we have addressed most of his/her concerns.

Minor points:

The new quantitation of the fibre assay data in 4C Fig can be confusing on its own as it can also indicate fork progression (with the second track being longer than the 1st track under treatment conditions). The authors should include quantitation of the length of the second track only to show that the track length is shorter following treatment compared to control, which is more representative of the images shown in Fig 4E

We thank the reviewer for his/her comment. We realized we had mislabeled the legend of the Y axis, that should read “IdU track length frequency”. This is likely the cause of the confusion. The quantification of the fiber assay data was presented in the original manuscript, and Fig.4C represents already the track length of the second analog (IdU), as explained in the Material and Methods section. If the reviewer refers to the new distribution dot plot shown in the insert of Fig 4C, this is not another representation of the fiber assay, but instead represents fork asymmetry. This analysis has been performed by measuring the ratio of the length of the second track in the two sides of individual bidirectional forks. We have clarified this point in the Material and Methods section.

Referee #3 (Report for Author)

The authors have addressed most of my previous questions and concerns. When possible, they also performed additional experiments to support their conclusions. Some of them (for example the replication fork asymmetry assay) allowed very interesting new insights. Unfortunately, a few important issues were not yet addressed adequately in my opinion. For example, alkaline comet assay is not appropriate to indicate the presence of DNA double-strand breaks, because when the strands are separated by the alkaline conditions, mostly single-strand nicks and gaps are revealed by this assay. Neutral comet assays should have been performed, as I suggested in my first review. Overall though, I think the authors did a decent job answering to the reviewers comments. Given the current international Corona crisis, where most labs (including my own) had to temporarily close down, I think it is a responsible position not to ask further experimental evidence from the authors, but only textual revisions. One such textual revision that I strongly suggest is the clear indication that the alkaline comet assay data in Fig 6F reveal predominantly single-strand nicks and gaps, but not double-strand breaks.

We thank the reviewer for his/her comments and are pleased he found that “some additional experiments allowed very interesting new insights”. As suggested, we state in the revised manuscript that the alkaline comet assay indicates predominantly single-strand nicks and gaps.

Accepted

14 April 2020

Thank you for sending in your final modifications. I am pleased to inform you that we have now accepted your manuscript for publication in The EMBO Journal.

Corresponding Author Name: George Thomas and Joffrey Pelletier

Journal Submitted to: Embo journal

Manuscript Number: EMBOJ-2019-103838R